# Beyond Formula Complexity: Effective Information Criterion Improves Performance and Interpretability for Symbolic Regression

## Abstract

Symbolic regression discovers accurate and interpretable formulas to describe given data, thereby providing scientific insights for domain experts and promoting scientific discovery. However, existing symbolic regression methods often use complexity metrics as a proxy for interoperability, which only considers the size of the formula but ignores its internal mathematical structure. Therefore, while they can discover formulas with compact forms, the discovered formulas often have structures that are difficult to analyze or interpret mathematically. In this work, inspired by the observation that physical formulas are typically numerically stable under limited calculation precision, we propose the Effective Information Criterion (EIC). It treats formulas as information processing systems with specific internal structures and identifies the unreasonable structure in them by the loss of significant digits or the amplification of rounding noise as data flows through the system. We find that this criterion reveals the gap between the structural rationality of models discovered by existing symbolic regression algorithms and real-world physical formulas. Combining EIC with various search-based symbolic regression algorithms improves their performance on the Pareto frontier and reduces the irrational structure in the results. Combining EIC with generative-based algorithms reduces the number of samples required for pre-training, improving sample efficiency by $2 \sim 4$ times. Finally, for different formulas with similar accuracy and complexity, EIC shows a $70.2\%$ agreement with 108 human experts' preferences for formula interpretability, demonstrating that EIC, by measuring the unreasonable structures in formulas, actually reflects the formula's interpretability. We provide code and data in `https://anonymous.4open.science/r/EIC-91B2`.

## 1 Introduction

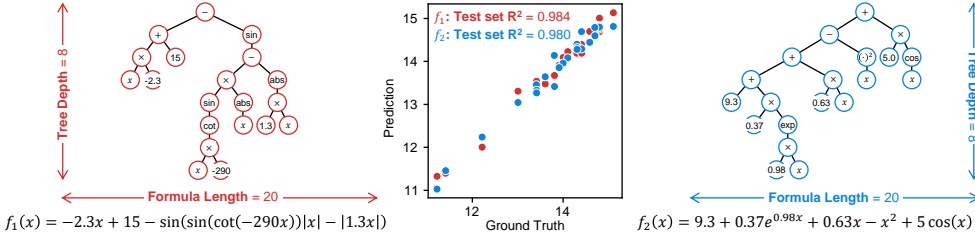

$$f_1(x) = -2.3x + 15 - \sin(\sin(\cot(-290x))|x| - |1.3x|)$$
$$f_2(x) = 9.3 + 0.37e^{0.98x} + 0.63x - x^2 + 5\cos(x)$$

Figure 1: **Formulas with the same complexity but different interpretability.** Formulas with similar complexity and accuracy can have very different structures and interoperability.

Symbolic regression (SR) is a machine learning technique that discovers interpretable mathematical formulas describing relationships in data (Camps-Valls et al., 2023). Unlike black-box models, it reveals how inputs map to outputs in a form that can be analyzed mathematically or logically, offering insights and new scientific knowledge (Makke & Chawla, 2024). In a typical workflow, researchers apply SR algorithms to **data** to generate candidate formulas that balance accuracy and complexity (i.e., the **Pareto front**). They can then assess the credibility of these candidates and select the

most interpretable **formula** for insights into the underlying patterns (Liu et al., 2024). Traditionally, SR relies on heuristic search, especially genetic programming, which edits formulas through mutation and crossover to maximize a search objective that balances complexity and accuracy (Cranmer, 2023; Burlacu et al., 2020). Recently, pretraining-based methods have gained attention, in which transformer models are pre-trained on synthetic formula-data pairs to predict formula symbols from data. After training, these models can generalize to real systems, generating formulas directly (Biggio et al., 2021; Kamienny et al., 2022; Meidani et al., 2023) or guiding symbolic search (Kamienny et al., 2023; Shojaee et al., 2023; Yu et al., 2025b).

Most SR methods use formula length, measured by the number of symbols in a formula (La Cava et al., 2021; Aldeia et al., 2025), as a proxy for interpretability. However, the assumption that shorter formulas are more interpretable is not always valid. The length metric only captures overall size but ignores the internal structure of a formula, which strongly affects interpretability. For example, although the two formulas in Figure 1 have the same length and similar accuracy, the left one is difficult to interprete with its nested structure $\sin(\sin(\cot(x)))$, whereas the right one, as a linear combination of simpler functions, is more likely to provide scientific insights. Thus, while search-based algorithms can discover short formulas, they cannot ensure interpretable structures, limiting the process of extracting knowledge and hindering SR in scientific discovery. On the other hand, generative methods face a similar issue: pre-trained on random formulas with inevitably unreasonable structures, they typically show poor sample efficiency and require tens or hundreds of millions of samples to achieve effective generalization (Kamienny et al., 2022; Yu et al., 2025b).

To address this problem, we propose the Effective Information Criterion (EIC), grounded in information theory and numerical analysis, to identify unreasonable structures in formulas. The key idea is to view a formula as an information-processing system and evaluate its structural plausibility based on the loss of effective information it outputs under finite processing precision, relative to the maximum information a system could output under the same precision. This approach is motivated by the observation that human-derived physical equations can typically be computed with limited precision (e.g., manual calculations or floating-point arithmetic) without substantial loss of significant digits. In contrast, formulas learned by SR often exhibit unreasonable structures such as catastrophic cancellation, leading to a large loss of significant digits under limited precision. As shown in Figure 2, while real physical formulas typically lose less than one significant digit, SR results, despite being compact and accurate, can lose three or more, revealing prevalent structural issues. Building on this, we integrate EIC into multiple stages of the SR workflow. For heuristic algorithms, it serves as an auxiliary search objective to guide the search toward more structurally reasonable results. For generative methods, EIC can filter out formulas with unlikely structures during pre-training, reducing the number of samples needed for effective generalization and thus improving sample efficiency.

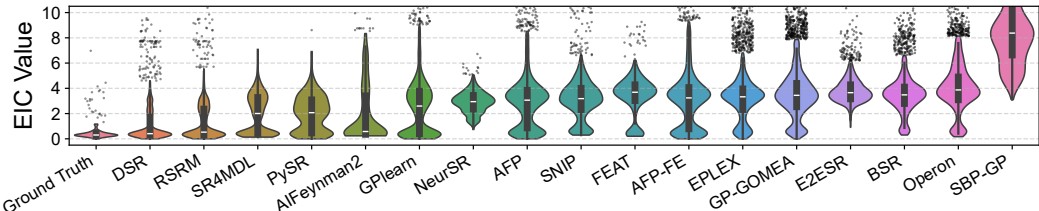

Figure 2: **EIC distributions of real physical formulas and existing SR results.** The figure compares the EIC distribution of 133 ground-truth physical formulas with that of formulas discovered by existing SR algorithms. While real physical formulas typically have a low EIC of $0 \sim 1$, formulas discovered by SR often exceed 3, highlighting the prevalence of unreasonable structures. The inner box shows the quartiles, and the black dots indicate outliers beyond $1.5 \times$IQR.

We conduct extensive experiments to demonstrate EIC's effectiveness in enhancing both the performance and interpretability of formulas learned by SR. First, unlike traditional complexity measures, EIC evaluates the plausibility of a formula's internal structure, providing a more accurate assessment of interpretability and potential for scientific insight. Second, integrating EIC into heuristic search algorithms enhances search performance, yielding formulas that occupy superior positions on the Pareto front with fewer unreasonable structures. Third, filtering high-EIC samples during pre-training for generative methods significantly reduces the number of training samples required, improving sample efficiency by 2 to 4 times. Models trained on low-EIC samples also generalize better to real-world physical formulas, achieving an $R^2$ improvement of 22.4% compared to those

trained on random formulas. Finally, in an evaluation of 172 formula pairs with similar $R^2$ and complexity but different EIC values, assessed by 108 human experts, we find that EIC aligns with human preferences 70.2% of the time. This demonstrates that EIC can not only identify unreasonable structures but also evaluate interpretability effectively. By incorporating EIC into the search, training, and evaluation stages of SR, we can enhance both algorithm performance and the potential to extract scientific knowledge from learned formulas.

## 2 RELATED WORK

Symbolic regression (SR) is a machine learning task that discovers symbolic formulas from data by combining variables, constants, and mathematical operators. Existing SR algorithms can be broadly divided into heuristic search-based and generative methods.

**Heuristic search-based symbolic regression methods.** Traditionally, SR is implemented using heuristic search methods such as genetic programming (Augusto & Barbosa, 2000; Schmidt & Lipson, 2010; de Franca & Aldeia, 2021; Arnaldo et al., 2014; Burlacu et al., 2020; Virgolin et al., 2019; Kartelj & Djukanović, 2023; Smits & Kotanchek, 2005; La Cava et al., 2016; Cranmer, 2023; Zhong et al., 2018; Burlacu et al., 2020; Searson et al., 2010; Virgolin et al., 2021; Zhang et al., 2022), Monte Carlo tree search (Sun et al., 2022), and deep reinforcement learning (Petersen et al., 2021; Tenachi et al., 2023). These methods search for functions that balance simplicity and accuracy, producing candidate formulas for experts' analysis. Some methods leverage neural networks to identify symmetries (Udrescu & Tegmark, 2020), fit Q-functions (Xu et al., 2024), or predict minimum description length (Yu et al., 2025b) to guide the search. While these methods often limit the maximal formula length to avoid overly complex formulas, they cannot prevent formulas with unreasonable structures, such as deeply nested nonlinear functions or catastrophic cancellation (i.e., subtracting nearly equal subformulas). Although there are methods using unit constraints to ensure physically reasonable structures (Tenachi et al., 2023), they are largely limited to physics. In broader domains like ecology, sociology, and earth science, where variable units may be unclear and models can include many constants with unknown units, such constraints are difficult to apply. This motivates incorporating EIC into the search objective to exclude unreasonably structured formulas, improving the interpretability of discovered results.

**Generative symbolic regression methods.** In recent years, generative methods have emerged, which pre-train transformers on large-scale randomly generated formula-data pairs to predict symbols in underlying formulas given data. Pre-trained models can generate formulas (Biggio et al., 2021; Kamienny et al., 2022), guide symbolic search (Shojaee et al., 2023; Kamienny et al., 2023; Yu et al., 2025b; Ying et al., 2025; Yu et al., 2025a), or serve as foundation models for downstream tasks (Meidani et al., 2023). Despite their promise, these methods face significant challenges in sample efficiency. Randomly generated formulas often contain unreasonable structures that differ from real-world physical formulas, requiring extremely large training datasets (tens or hundreds of millions) for effective generalization. This motivates using EIC to filter out formulas with unreasonable structures during pre-training, thereby improving consistency with real-world formulas and enhancing sample efficiency.

## 3 METHODOLOGY

In this section, we introduce the proposed Effective Information Criterion (EIC), illustrated in Figure 3. Section 3.1, corresponding to the upper part of the figure, introduces its definition and physical meanings from perspectives of numerical analysis and information-processing systems. Section 3.2, corresponding to the lower part, introduces three ways EIC can enhance symbolic regression workflows and facilitate scientific discovery.

### 3.1 EFFECTIVE INFORMATION CRITERION (EIC)

**Definition.** We view a symbolic formula not merely as a static mathematical object, but as an information-processing system. A formula maps inputs to outputs through a sequence of mathematical operations and thus forms an information flow. Under limited processing precision, rounding errors inevitably occur at operator nodes and propagate through the computation graph. In

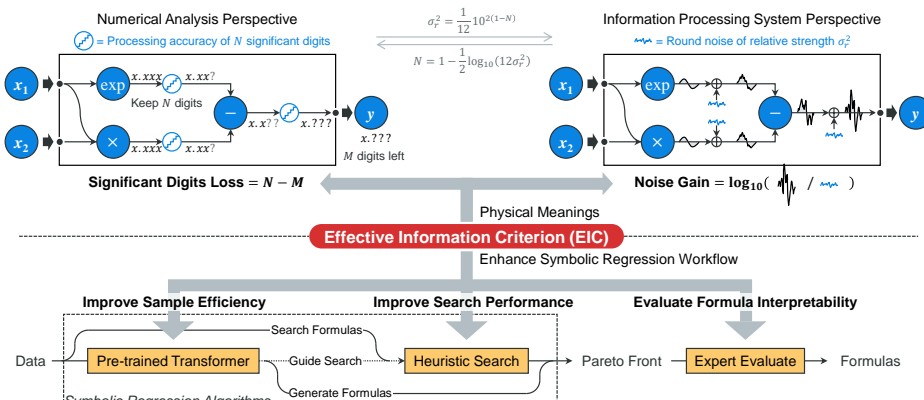

Figure 3: **Demonstration of the physical meanings of EIC and the way it can enhance symbolic regression workflows.**

well-structured formulas, these errors remain controlled, so the effective information at the output approaches the system's maximum. In unreasonably structured formulas, however, small intermediate perturbations can accumulate and severely degrade the output's effective information. This motivates evaluating a formula's structure by how stably it processes information and how much effective information is lost under finite precision.

To quantify this, we introduce the Effective Information Criterion (EIC), which measures information loss in terms of significant digits. If a formula computed with $N$ significant digits produces an output with only $M$ significant digits, its EIC is defined as

$$\text{EIC} \triangleq N - M. \tag{1}$$

Therefore, a large EIC indicates a formula that amplifies rounding errors and leads to severe information loss, while a small EIC indicates that the formula structure preserves most of the output information. For example, $f(x) = (x + 10^{100}) - 10^{100}$ suffers catastrophic cancellation when computed with fewer than 100 significant digits, leaving no reliable information output. In contrast, physical formulas from science typically preserve output information with finite precision, resulting in small EIC values.

**Calculation.** Limited processing precision can be modeled as noise injected into intermediate computations. Specifically, truncating a variable $x$ to $N$ significant digits can be expressed as $\tilde{x} = x + e$, where the error term $e$ behaves like uniform noise following $\mathcal{U}(-5 \times 10^{M-N}, 5 \times 10^{M-N})$, with $M = \log_{10}|x|$ representing the order of magnitude of $x$. Expressed as relative noise $e/x$, it intensity is independent of $x$ (and $M$):

$$\sigma_r^2 \triangleq \text{Var}\left[\frac{e}{x}\right] = \frac{1}{12} 10^{2(1-N)}, \tag{2}$$

or equivalently,

$$N = 1 - \frac{1}{2} \log_{10}(12\sigma_r^2). \tag{3}$$

This shows that retaining a certain number of significant digits can be converted into injecting noise of a specific intensity. This equivalence provides us a practical method for calculating EIC: we first compute the formula outputs $\tilde{y}$ with noise of relative strength $\sigma_r^2$ added. Then, we estimate the relative noise strength at the output through $\delta_r^2 \triangleq \text{Var}\left[\frac{\tilde{y}-y}{y}\right]$, where $y$ is the noiseless output. Finally, the output noise can be converted back to significant digits as $M = 1 - \frac{1}{2} \log_{10}(12\delta_r^2)$, using the relationship above.

**Physical meanings.** The correspondence between significant digits and relative noise intensity provides both a method for calculating EIC and a richer physical interpretation. Specifically, from equation 3, we have (see Appendix B.1 for details)

$$\text{EIC} = N - M = \log_{10}(\delta_r^2/\sigma_r^2). \tag{4}$$

This highlights EIC's dual meaning: from a numerical analysis perspective, it measures the number of significant digits lost (i.e., $N - M$), while from an information-processing perspective, it quantifies the relative noise gain along the formula's computational graph (i.e., $\log_{10}(\delta_r^2/\sigma_r^2)$). In other

words, EIC captures the sensitivity of a formula's output to small intermediate perturbations, thus identifying formulas that may be compact but mathematically fragile. Both perspectives emphasize EIC's value as a physically meaningful criterion for detecting unreasonable mathematical structures.

**Implementaion details.** EIC can be calculated recursively. As shown in Appendix Algorithm 1, computing a formula's EIC requires computing the noisy and noise-free outputs of each subformula first, which further depend on their subformulas, down to the input variables and constants. This allows EIC to be computed for both the final output and each subformula. We found that in some cases, subformulas have higher EICs than the full formula. For example, $f(x) = 0 \times (x + 10^{100} - 10^{100}) + x$ contains catastrophic cancellation inside the parentheses, but the zero coefficient nullifies its effect, resulting in a small EIC at the output. To account for such cases, we use the maximum EIC among all subformulas as the formula's overall EIC.

### 3.2 ENHANCE SYMBOLIC REGRESSION WITH EIC

**Improve heuristic search-based methods.** EIC evaluates unreasonable structures in formulas and can be integrated as an auxiliary objective in classical heuristic search algorithms to guide the search and improve performance. We focus on two representative approaches: genetic programming (GP) and Monte Carlo tree search (MCTS), which underpin many state-of-the-art methods such as PySR(Cranmer, 2023), TPSR(Shojaee et al., 2023), SR4MDL(Yu et al., 2025b), etc. GP and MCTS iteratively edit formulas to maximize a search objective, balancing accuracy and complexity. A commonly used fitness function is defined as

$$f(C, \text{NMSE}; \eta) = \eta^C / (1 + \text{NMSE}), \tag{5}$$

where $C$ denotes the formula complexity (length), $\text{NMSE} = \text{MSE}/\text{Var}(y)$, and $\eta < 1$ penalizes formula complexity (Sun et al., 2022) (see Appendix B.2 for details). It is worth noting that, when calculating the MSE, we first decompose formulas into additive terms and then fit their coefficients using linear regression, avoiding more costly nonlinear optimization algorithms such as BFGS(Fletcher, 2000) and accelerating the search. EIC can be incorporated by modifying the search objectives:

$$f(C, \text{NMSE}, \text{EIC}; \eta, \alpha) = \eta^{\text{Complexity}} / (1 + \text{NMSE}) - \alpha \cdot \text{EIC}, \tag{6}$$

where $\alpha > 0$ penalizes formulas with higher EIC, steering the search away from structurally unreasonable solutions.

**Improve sample efficiency of generative methods.** Generative approaches, such as NeuralSR(Biggio et al., 2021), E2ESR (Kamienny et al., 2022), and SNIP (Meidani et al., 2023), are typically pretrained on synthetic formula-data pairs, aiming to predict tokens (i.e., symbols) in formulas from input data. These methods often rely on the random formula generation algorithm developed by Lample & Charton (2020), which produces diverse symbolic expressions for pretraining. While this enables generating unlimited training data for pre-training, many synthetic formulas contain mathematically unreasonable structures rarely seen in real-world physical equations. Therefore, these models often require tens to hundreds of millions of samples and weeks of computation for effective generalization(Kamienny et al., 2022; Meidani et al., 2023). To address this, we propose filtering the pretraining corpus via rejection sampling based on EIC. By discarding formulas with high EIC values, we remove structurally implausible samples while preserving overall diversity. Specifically, during pretraining, we evaluate the EIC value of each generated sample. Samples exceeding a specified threshold are regenerated until the EIC is below the threshold.

## 4 EXPERIMENTS

In this section, we demonstrate practical applications of the Effective Information Criterion (EIC) in symbolic regression, including: 1) identifying prevalent unreasonable structures in existing SR results, 2) enhancing the performance of various heuristic search-based methods, 3) improving the sample efficiency of various generative methods, and 4) evaluating formula interpretability in alignment with domain expert intuition.

### 4.1 EIC EVALUATES EXISTING SYMBOLIC REGRESSION METHODS

**Experimental setups.** To assess the prevalence of unreasonable structures in existing SR outputs, we compare the EIC values of 17 symbolic regression methods on 133 SRbench white-box problems

against the ground-truth formulas. The SRbench set includes 119 Feynman physics formulas and 14 Strogatz ODE formulas, with corresponding synthetic data. Samples are randomly split into 75% training data for search and 25% test data for evaluation. Each method is run 10 times per problem, and results with $R^2 > 0.8$ are retrained, excluding poorly fitting formulas. This filtering removes approximately 39% of the data points.

**Finding 1. Gap with real-world formulas in terms of EIC.** Figure 4 shows the average EIC values and complexities (formula lengths) of the evaluated algorithms alongside the 133 ground-truth formulas. Although most methods can discover compact formulas comparable in complexity to the ground truth, their EIC values are consistently higher, revealing widespread unreasonable structures and highlighting the need to incorporate EIC into evaluation.

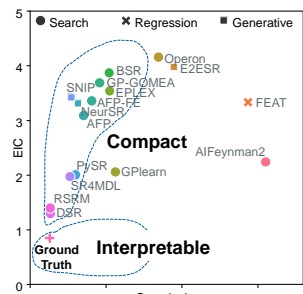

Figure 4: **Comparison of complexity and EIC between ground-truth physical formulas and symbolic regression results.**

**Finding 2. Unit constraints and hand-designed rules improve structural plausibility.** Among all methods, DSR achieves the closest EIC to the ground-truth, likely due to its use of dimensional constraints, which prevent nonlinear unary operators such as $exp$ and $sin$ from being applied to dimensional subformulas, thereby pruning the search space and avoiding overly complex structures. PySR and GPlearn follow it with suboptimal EIC values, possibly because they restrict deep nesting of nonlinear unary functions through hand-designed rules, thus reducing structurally complex results.

**Finding 3. Divide-and-conquer improves structural pausibility.** AIFeynman2 and SR4MDL achieve performance similar to PySR and GPlearn, likely because they use a divide-and-conquer strategy that breaks the target formula into simpler subformulas, reducing abnormal structures. Notably, although AIFeynman2 produces formulas of high average complexity, their EIC values remain relatively low. This is likely due to its design, which uses a series of pre-defined symmetry rules to recursively rewrite formulas into smaller components. Since each divide-and-conquer step stems from simple symmetry rules, it avoids the unreasonable structures with high EIC values.

**Finding 4. Generative methods tend to produce structurally unreasonable formulas.** Compared to search-based methods, recent generative methods such as NeurSR, E2ESR, and SNIP yield formulas with much higher EIC values. This likely stems from training on randomly generated formulas that ignore structural plausibility, making them reproduce these abnormal substructures.

These findings highlight the effective practices in existing symbolic regression methods, while also revealing a gap between the structural plausibility of generated formulas and that of truly interpretable symbolic models, motivating our exploration of EIC to improve existing SR algorithms in the following experiments.

## 4.2 Incorporating EIC into Heuristic Search Methods

**Experimental setups** We evaluated GP and MCTS with and without EIC integrated on SRBench, which is a standard benchmark for SR, including 133 white-box problems with known ground-truth formulas and 122 black-box problems without explicit underlying dynamics. For each problem, we ran 10 independent trials with a four-hour time limit and compared results against the 17 baseline methods in Section 4.1. Data were split into 75% training and 25% test sets. For white-box problems, we tested both noise-free data and noisy data with three levels $\eta = 0.001, 0.01, 0.1$, where Gaussian noise with standard deviation $\sigma = \eta \times \text{Std}[y]$ was added to the outputs, allowing us to assess robustness under varying noise. Performances of algorithms are evaluated from two complementary perspectives, including 1) the trade-off between accuracy ($R^2$) and complexity (formula length) that reflects the ability of the algorithm to learn compact yet accurate models, and 2) the average EIC values of discovered formulas that reflect their structural plausibility.

**White-box experimental results.** Figure 5 summarizes the white-box results. As shown in the left panel, integrating EIC into MCTS and GP improved both accuracy and compactness, moving them from the second to the first tier of the Pareto front. This is because EIC, as a regularizer, can prevent overfitting and enhance generalization accuracy to test set data. It also reduces formula length by penalizing unnecessary complexity. In contrast, while methods such as DSR and RSRM produce compact formulas with relatively low EIC values (Figure 4), their predictive accuracy is much lower.

This suggests that although unit constraints can avoid unreasonable structures, they also shrink the search space and limit expressiveness

The right panel shows the EIC distribution under different noise levels. With EIC, both MCTS and GP consistently achieve lower values than their vanilla versions, approaching those of real physical formulas. This demonstrates that EIC not only improves performance but also prevents the generation of structurally unreasonable, hard-to-interpret formulas, making it a better constraint solution than unit constraints or manually designed nested rules.

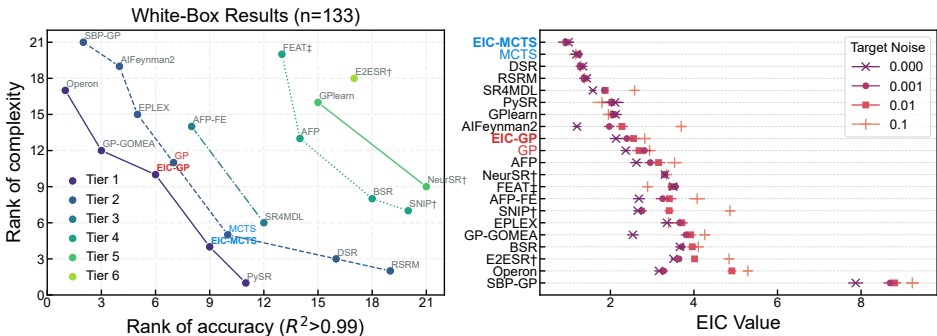

Figure 5: **Pareto fronts and EIC distributions on the white-box dataset.** † and ‡ indicate generative and regression methods, respectively, while others are heuristic methods. In the left panel, the lines show the Pareto front tiers, from bottom-left (best) to top-right (worst). In the right panel, markers indicate the mean EIC and its 95% confidence interval at different noise levels.

**Black-box experimental results.** Figure 6 shows the results on 122 black-box problems. In the left panel, incorporating EIC improved the Pareto front positions of GP and MCTS, moving them from Tier 2 and Tier 4 to Tier 1 and Tier 3, respectively, demonstrating that excluding structurally unreasonable formulas with EIC can enhance symbolic regression performance even on real-world problems without known ground truth. However, while EIC improved both accuracy and simplicity for MCTS, it improved simplicity but reduced accuracy for GP. This likely reflects the presence of datasets with relationships too complex to be captured by interpretable formulas of reasonable structure. As a regularizer, EIC guides search away from unreasonable forms, but also prevents generating overly complex formulas to overfit the intricate patterns in the data.

The right panel shows EIC distributions, where incorporating EIC substantially reduces unreasonable structures in both MCTS and GP. Compared to the white-box results in Figure 5, this effect is even stronger in the black-box setting, suggesting that when underlying relationships are unclear, symbolic regression tends to rely on overly complex functional forms, leaving more room for improvement in formula plausibility.

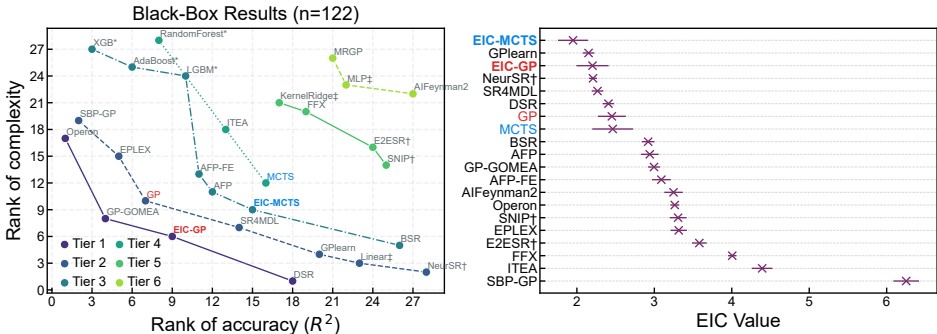

Figure 6: **Pareto fronts and EIC distributions on the black-box dataset.** ∗ indicates decision-tree methods, other symbols are the same as in Fig 5.

### 4.3 Incorporate EIC into Generative Symbolic Regression Methods

**Experimental setups.** In this experiment, we considered E2ESR (Kamienny et al., 2022), SNIP (Meidani et al., 2023), and SR4MDL (Yu et al., 2025b), covering the progression of this methodology from earlier efforts to more recent advances over the past two years. Each method

was trained on randomly generated formulas and evaluated on the 133 SRBench white-box problems. For E2ESR and SNIP, we used the $R^2$ of generated formulas as the performance metric; for SR4MDL, which predicts minimum description length rather than formula tokens[1], we used the RMSE and MAE of its prediction as metrics. Training stopped when test performance plateaued over five million samples. We then retrain the models under the same hyperparameters using filtered formulas with EIC $< \theta$ until performance matches that obtained with unfiltered data. Based on Figure 2, we set $\theta = 2$ to align the EIC of filtered formulas with that of real physical formulas (see Appendix B.3 for details).

We also evaluate a recent data construction baseline, PhyE2E (Ying et al., 2025), which uses LLMs fine-tuned on physical equations to generate "look-physical" formulas with unit constraints for pretraining. We trained the E2ESR model on 180k PhyE2E formulas. To avoid unfair comparisons due to its limited sample size, we used a mixed sampling strategy that samples from PhyE2E formulas with a probability of 0.1 and generates random formulas with a probability of 0.9.

**EIC reduces the distribution gap between training samples and real physical formulas.** We evaluated whether EIC filtering produces a training distribution closer to real-world formulas. We generated 1024 formulas from the unfiltered random generator and 1024 from the filtered generator, and compared them with three benchmark sets, including Feynman (119 physics formulas), Strogatz (14 ODE formulas), and Wiki Named Equations (984 equations, see Appendix C.2 for details). Similarity was measured via variable counts, constant counts, operator counts, and formula length, using Jensen-Shannon (JS) and Kullback-Leibler (KL) divergences.

Table 1 shows that EIC filtering yields distributions substantially closer to real formulas, with $20 \sim 50\%$ reduction in JS divergence and $30 \sim 70\%$ reduction in KL divergence compared to the unfiltered random formulas. This demonstrates that EIC effectively reduces the gap between synthetic and real-world formulas and improves train-test alignment.

**EIC improves sample efficiency and generalization performance.** The results are summarized in Table 2, where models trained on randomly generated formulas eventually generalized to the Feynman test set, but typically required tens of millions of samples. In contrast, training on EIC-filtered formulas achieved equal or better performance with fewer training samples, improving sample efficiency by 357%, 233%, and 287% on E2ESR, SNIP, and SR4MDL, respectively. This demonstrates that EIC filtering removes structurally unreasonable formulas, producing training data that is more physically meaningful and transferable to real-world tasks, thereby enhancing pretraining-based symbolic regression.

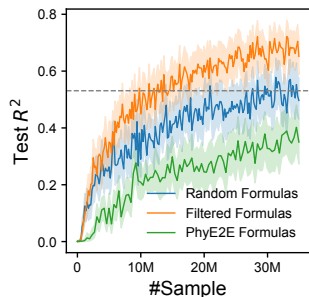

Figure 7: **Test set performance of E2ESR trained on different samples.** The grey line shows the convergence trained on random formulas.

Figure 7 shows that training on EIC-filtered samples further improved final performance compared to random formulas. For E2ESR, training for the same number of steps increased $R^2$ from 0.5559 to 0.6808, representing a 22.4% relative gain. Similar improvements were observed for SNIP and SR4MDL, with relative gains of 13.5% and 5.14%, respectively (see Appendix C.2). Incorporating PhyE2E formulas into pretraining did not improve performance and slightly reduced generalization, likely due to a trade-off between unit consistency and formula diversity: although PhyE2E formulas satisfy unit constraints, their limited diversity reduces pretraining effectiveness. In contrast, EIC filtering preserves both structural plausibility and distributional diversity, thus yielding superior generalization.

### 4.4 EIC REFLECTS INTERPRETABILITY IN ALIGNMENT WITH HUMAN EXPERTS

**Experimental setups.** To test whether EIC aligns with human judgments of formula interpretability, we compared formulas discovered by EIC-MCTS and PySR, which were selected for their top performance in EIC and complexity, respectively (Section 4.2). We focused on Pareto fronts they discovered from 19 one- and two-dimensional PMLB black-box problems, where data can be visualized and understood by human experts. From each problem, we selected pairs of formulas with

---

[1]Strictly speaking, this is not a generative method. However, since it uses the same model architecture and data generation scheme as E2ESR and SNIP, we include it for comparison.

Table 1: **Distribution differences between real physical formulas and generated formulas.** We generated 1024 random formulas and 1024 EIC-filtered formulas (EIC $< 2.0$) and compared them with three real physical formula sets.

| | | #Variables | | #Coefficients | | #Operators | | Formula Length | |
|---|---|---|---|---|---|---|---|---|---|
| | | JS↓ | KL↓ | JS↓ | KL↓ | JS↓ | KL↓ | JS↓ | KL↓ |
| **Feynman** | Random | 0.1196 | 1.4390 | 0.5163 | 16.78 | 0.3233 | 9.539 | 0.3973 | 13.615 |
| (n=119) | Filtered | 0.0768 | 0.3523 | 0.3769 | 5.582 | 0.1862 | 0.7741 | 0.2573 | 3.4397 |
| | $-\Delta(\%)$ | **36%** | **76%** | **27%** | **67%** | **42%** | **92%** | **35%** | **75%** |
| **Strogatz** | Random | 0.4014 | 18.784 | 0.6016 | 23.37 | 0.5726 | 22.24 | 0.6002 | 22.27 |
| (n=14) | Filtered | 0.2650 | 12.920 | 0.5048 | 20.78 | 0.4646 | 18.74 | 0.5225 | 20.05 |
| | $-\Delta(\%)$ | **34%** | **31%** | **16%** | **11%** | **19%** | **16%** | **13%** | **10%** |
| **Wiki** | Random | 0.1459 | 0.6654 | 0.5566 | 14.888 | 0.3873 | 4.269 | 0.4219 | 7.060 |
| (n=940) | Filtered | 0.0747 | 0.2899 | 0.4660 | 3.5307 | 0.2485 | 1.124 | 0.3114 | 1.541 |
| | $-\Delta(\%)$ | **49%** | **56%** | **16%** | **76%** | **36%** | **74%** | **26%** | **78%** |

Table 2: **Sample efficiency of different methods trained with random and filtered formulas.**

| Pre-training Formulas | E2ESR | | SNIP | | SR4MDL | | |
|---|---|---|---|---|---|---|---|
| | #Sample↓ | R2↑ | #Sample↓ | R2↑ | #Sample↓ | MAE↓ | RMSE↓ |
| Random | 35M | 0.5190 | 40M | 0.5300 | 50M | 6.972 | 8.731 |
| Filtered | **9.79M** | **0.5399** | **25.15M** | **0.5415** | **17.4M** | **6.812** | **8.701** |
| Efficiency $+\Delta(\%)$ | 357.5% | | 198.8% | | 287.4% | | |

similar $R^2$ and complexity but differing EIC, yielding 172 pairs across all problems. We invite 108 volunteer participants with at least a bachelor's degree in science or engineering, to whom we assign 10 pairs of formulas and ask to choose the more interpretable ones from each pair. We collected 840 valid evaluations, with each pair assessed on average 4.9 times (see Appendix C.3).

**EIC evaluates formula interpretability in alignment with human experts.** The results are summarized in Figure 14, where the left two panels show the distribution of complexity and $R^2$ for the selected formula pairs, indicating comparable performance between the two methods. The right two panels show that EIC-MCTS produces formulas with substantially lower EIC values than PySR, and human experts preferred EIC-MCTS in 70.12% of evaluations, which is 2.3 times more than PySR. This suggests EIC captures interpretability in a manner aligned with expert intuition. Moreover, when asking LLMs to act as domain experts, they similarly preferred EIC-MCTS 72.19% of the time, supporting the reliability of the rating results (see Appendix C.3).

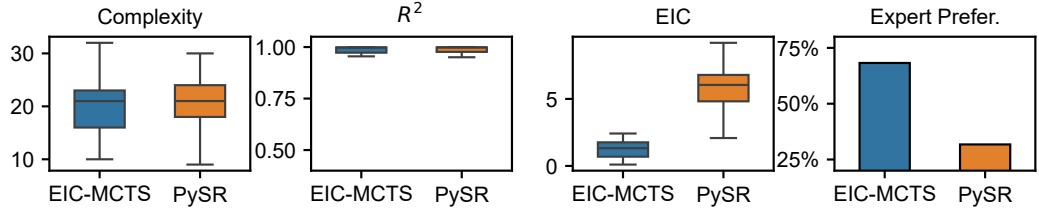

Figure 8: **Distributions of complexity, accuracy, EIC, and expert preference of formula pairs.**

## 5 CONCLUSION AND DISCUSSION

We propose the Effective Information Criterion (EIC), a measure of structural plausibility for symbolic regression formulas with interpretations from numerical analysis and information-processing perspectives. Unlike complexity-based metrics, EIC detects unreasonable structures that affect interpretability. It improves heuristic search-based SR by steering away from unreasonable formulas, enhances generative methods by filtering implausible samples, and aligns well with expert judgments for interpretability. Despite its effectiveness, several avenues remain for future work. First, integrating EIC guidance with pre-trained model guidance may further boost performance. Second, analyzing EIC at each subformula, rather than only the maximum, could more precisely identify redundant or unreasonable structures, enabling finer-grained evaluation. Finally, in complex systems where simple microscopic rules produce emergent macroscopic behaviors, such as phase transitions, it remains to be studied whether the resulting dynamics formulas retain low EIC scores.

## ETHICS STATEMENT

This research included an expert rating experiment with 100 participants, who were asked to evaluate the interpretability of 10 pairs of discovered symbolic formulas. Participation was entirely voluntary, and informed consent was obtained from all participants before the study. No personally identifiable or sensitive data were collected, and all responses were anonymized. The study was conducted in accordance with ethical research practices and posed no foreseeable risks to participants.

## REPRODUCIBILITY STATEMENT

To ensure reproducibility, we provide a complete implementation in an anonymous repository, including 1) scripts for computing the EIC score of any given formulas, 2) full implementations of the genetic programming and Monte Carlo Tree Search algorithms combined with EIC, and 3) full implementations of filtering samples with EIC to pre-train generative models. The repository also contains all datasets used in our experiments, including 1) the formula collections and EIC scores of the Feynman, Strogatz ODE, and Wiki Named Equation sets, 2) the EIC scores of formulas discovered by different symbolic regression methods, and 3) the detailed results of the human evaluation experiments.

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

# A    DISCLOSURE OF LLM USAGE

We used a large language model (LLM) as a general-purpose assistive tool. Specifically, we use GPT4-mini for writing assistance, including polishing text, correcting grammar, and improving phrasing. We also use Copilot for code documentation, that is, adding explanatory comments to source code for readability. All content generated by LLM is manually reviewed and edited to ensure correctness. The LLM did not contribute to research ideation, experiment design, implementation of experimental code, analysis of results, or drawing of conclusions. All scientific contributions and research insights presented in this paper are solely due to the authors.

# B    METHODOLOGY DETAILS

## B.1    EFFECTIVE INFORMATION CRITERION

Using equation 2 and equation 3, we can derive that:

$$
\begin{aligned}
\mathrm{EIC} &= N - M \\
&= 1 - \frac{1}{2}\log_{10}(12\sigma_r^2) - 1 + \frac{1}{2}\log_{10}(12\delta_r^2) \\
&= \log_{10}(\delta_r^2/\sigma_r^2),
\end{aligned}
\tag{7}
$$

demonstrating the physical meaning of EIC from the perspective of the information-processing systems. Based on this equation, we give the calculation algorithm of EIC as in Algorithm 1.

In this research, we kept the hyperparameter $\sigma_r$ as $10^{-6}$. We also find that, when the $\sigma_r$ is small enough, the value of EIC is independent of the specific value of $\sigma_r^2$ or $N$. This actually makes our EIC a parameter-free metric, reflecting the intrinsic plausibility of a formula independent of external tuning.

## B.2    ENHANCE SEARCH-BASED METHODS WITH EIC

To demonstrate that EIC, as a criterion for evaluating unreasonable structures in formulas, can serve as an effective guidance signal for improving symbolic regression, we integrated it into various classical heuristic search algorithms as an auxiliary search objective. Specifically, we focused on two representative approaches: genetic programming (GP) and Monte Carlo tree search (MCTS).

- Genetic programming maintains a population of candidate formulas, where each individual represents a candidate formula. New individuals are generated by crossover or mutation of

---

**Algorithm 1:** CalculateEIC

---

**Input:** Formula $f$, input data $X$, hyperparameter $\sigma_r^2$ ($\sigma_r^2 \to 0$)
**Output:** Noisy output $\tilde{y}$, clean output $y$, EIC value

1 **if** *f is an unary operator* **then**
2     operator $\leftarrow f.$operator ;             `// operator` $\in \{\sin, \cos, \tan, \exp, \log, \cdots\}$
3     operand $\leftarrow f.$operand;
4     $(\tilde{x}, x, \text{EIC}) \leftarrow \text{CalculateEIC}(\text{operand}, X)$;
5     $y \leftarrow \text{operator}(x)$;
6     $\tilde{y} \leftarrow \text{operator}(\tilde{x})$;
7     $\tilde{y} \leftarrow \tilde{y} + \epsilon\tilde{y}, \; \epsilon \sim \mathcal{N}(0, \sigma_r^2)$;
8     $\delta_r^2 \leftarrow \text{Var}\left[\frac{\tilde{y}-y}{y}\right]$;
9     $\text{EIC} \leftarrow \max\left\{\text{EIC}, \log_{10}(\delta_r^2/\sigma_r^2)\right\}$;
10     **return** $(\tilde{y}, y, EIC)$;
11 **else if** *f is a binary operator* **then**
12     operator $\leftarrow f.$operator ;             `// operator` $\in \{+, -, \times, \div, \cdots\}$
13     $\text{operand}_1 \leftarrow f.\text{operand}[1]$;
14     $\text{operand}_2 \leftarrow f.\text{operand}[2]$;
15     $(\tilde{x}_1, x_1, \text{EIC}_1) \leftarrow \text{CalculateEIC}(\text{operand}_1, X)$;
16     $(\tilde{x}_2, x_2, \text{EIC}_2) \leftarrow \text{CalculateEIC}(\text{operand}_2, X)$;
17     $y \leftarrow f(x_1, x_2)$;
18     $\tilde{y} \leftarrow f(\tilde{x}_1, \tilde{x}_2)$;
19     $\tilde{y} \leftarrow \tilde{y} + \epsilon\tilde{y}, \; \epsilon \sim \mathcal{N}(0, \sigma_r^2)$;
20     $\delta_r^2 \leftarrow \text{Var}\left[\frac{\tilde{y}-y}{y}\right]$;
21     $\text{EIC} \leftarrow \max\left\{\text{EIC}_1, \text{EIC}_2, \log_{10}(\delta_r^2/\sigma_r^2)\right\}$;
22     **return** $(\tilde{y}, y, EIC)$;
23 **else**
24     **return** $(f(X), f(X), 0)$ ;          `// f is a variable or a constant`

---

high-fitness candidates, while low-fitness candidates are eliminated to increase the overall quality of the population. A common fitness function is defined in equation 5, where *Complexity* measures the structural size of the formula, NMSE denotes the normalized mean squared error (MSE/Var($y$)), and $\eta < 1$ is a regularization constant. This formulation penalizes formulas with large errors or excessive complexity, thereby guiding the search toward accurate and compact formulas. It is worth noting that, when calculating the MSE, we first decompose formulas into additive terms and then apply linear regression to fit the coefficients for these terms. This approach integrates the idea of SINDy and effectively improves the performance of algorithms.

- MCTS constructs a search tree where each node represents a candidate formula. Child nodes are those formulas that can be obtained by mutating parent formulas, and their reward values are computed according to Equation equation 5. In each search iteration, the algorithm starts from the root node and selects a promising leaf node based on its upper confidence bound (UCB) score, which balances average reward and visitation counts. The chosen leaf is then expanded through mutation, and the resulting reward is backpropagated to update the average reward and visitation counts of its ancestors. This process iteratively guides the search toward structurally and numerically promising formulas.

The proposed EIC can be easily incorporated into both algorithms by augmenting their fitness or reward functions. Specifically, we adopt

$$\text{Fitness}_\alpha = \eta^{\text{Complexity}}/(1 + \text{NMSE}) - \alpha \cdot \text{EIC}$$

as the modified fitness and reward functions, where $\alpha > 0$ penalizes formulas with higher EIC, thereby steering the search away from structurally unreasonable solutions. In this work, we use $\eta = 0.999$ in equation 5 as suggested by Yu et al. (2025b) and Sun et al. (2022). For the choice of $\alpha$, we note that the first term of equation 5 has a value range of 0 to 1, while, as shown in Figure 2, EIC ranges from 0 to 10. Since EIC serves as an auxiliary search objective, its weight should generally be an order of magnitude smaller than that of the primary objective. Accordingly, we set $\alpha = 0.01$ for MCTS. For the genetic algorithm, we observed that it is more sensitive to the auxiliary objective. Therefore, we chose a smaller value of $\alpha = 0.002$ to optimally balance the algorithm's ability to discover formulas with low EIC while still ensuring high $R^2$ accuracy.

### B.3 ENHANCE GENERATIVE METHODS WITH EIC

To demonstrate that EIC can enhance the sample efficiency and performance of pretraining-based symbolic regression methods, we focus on three representative approaches: E2ESR(Kamienny et al., 2022), SNIP(Meidani et al., 2023), and SR4MDL(Yu et al., 2025b), spanning the progression of this research line from earlier efforts to more recent advances in the last year. E2ESR is trained to predict formula tokens directly from $(X, y)$ data pairs; SNIP extends E2ESR with a CLIP-inspired contrastive loss; and SR4MDL further modifies SNIP by changing the prediction target to the minimum description length of formulas to guide the search. All three approaches rely on the formula generation algorithm proposed by Lample & Charton (2019), which generates formulas with random and diverse forms to pretrain generative models.

For each method, we pretrained the model on randomly generated formulas until it achieved the performance reported in the corresponding original papers, and then repeated the pretraining process using a filtered dataset where formulas with EIC > 2.0 are discarded, ensuring that only formulas with reasonable structures were included in pre-training. For E2ESR and SNIP, since they directly predict formula tokens from data, we evaluated their performance by the $R^2$ scores of the formulas they generated on the Feynman dataset. For SR4MDL, which instead predicts the minimum description length of formulas, we evaluated its performance using the mean absolute error (MAE) and root mean squared error (RMSE) of predicted formula lengths on the Feynman dataset. This setup allowed us to compare the number of training samples required to reach the same target performance under both settings. All experiments used their official implementations, with hyperparameters selected via a grid search over batch sizes $\in \{32, 64, 128, 256\}$ and learning rates $\in \{10^{-5}, 10^{-4}, 10^{-3}\}$. The hyperparameters were selected under the unfiltered training condition and then kept fixed when training with the filtered dataset.

We also considered a recently proposed data construction strategy baseline, PhyE2E(Ying et al., 2025), which leverages LLMs fine-tuned on physical equations to generate "look-physical" formu-

las with unit constraints for pretraining generative models. We used 180k formulas provided by PhyE2E to train the model. To avoid unfair comparisons due to limited sample size, we used a mixed sampling strategy, which is to generate random formulas with a probability of 0.9 and sample from PhyE2E formulas with a probability of 0.1. We used the 180k formulas provided by PhyE2E to pretrain the model and compared the results against those obtained with the EIC-filtered dataset. To ensure a fair comparison with our approach and to mitigate the limited size of the PhyE2E corpus, we adopted a hybrid sampling scheme in which, at each training step, a formula was drawn from the PhyE2E dataset with probability 0.1 and from the model's own random generator with probability 0.9.

## C  Detailed Experimental Results

### C.1  Enhance Search-based Methods with EIC

Corresponding to Figures 5 and 6 in the main text, we provide the raw experimental results for 17 baseline methods and our four methods under four noise levels on white-box data and on black-box data, as shown in Tables 3, 4, 5, 6, and 7, respectively. The tables report formula accuracy, complexity, running duration, and the average EIC of the resulting formulas.

Table 3: Whitebox results at noise-free condition

| Type | algorithm | $R^2 > 0.99$ | complexity | duration | EIC |
|---|---|---|---|---|---|
| Regression | **FEAT** | $0.621_{(\pm0.031)}$ | $195.3_{(\pm8.5)}$ | $2269_{(\pm1.9e+02)}$ | $3.504_{(\pm0.22)}$ |
| Generative | **E2ESR** | $0.2773_{(\pm0.03)}$ | $89.63_{(\pm2.1)}$ | $4.024_{(\pm0.14)}$ | $3.509_{(\pm0.066)}$ |
| | **NeurSR** | $0.07681_{(\pm0.014)}$ | $31.42_{(\pm0.26)}$ | $23.19_{(\pm0.39)}$ | $3.307_{(\pm0.049)}$ |
| | **SNIP** | $0.1541_{(\pm0.019)}$ | $25.2_{(\pm0.39)}$ | $1.845_{(\pm0.07)}$ | $2.652_{(\pm0.1)}$ |
| Search | **AFP** | $0.551_{(\pm0.03)}$ | $37.01_{(\pm1.2)}$ | $3282_{(\pm2.4e+02)}$ | $2.622_{(\pm0.15)}$ |
| | **AFP-FE** | $0.7154_{(\pm0.025)}$ | $40.63_{(\pm1.3)}$ | $17090_{(\pm750)}$ | $2.684_{(\pm0.14)}$ |
| | **AIFeynman2** | $0.8487_{(\pm0.02)}$ | $113.2_{(\pm21)}$ | $844.2_{(\pm1.9e+02)}$ | $1.2_{(\pm0.2)}$ |
| | **BSR** | $0.249_{(\pm0.026)}$ | $26.95_{(\pm0.62)}$ | $29310_{(\pm790)}$ | $3.663_{(\pm0.11)}$ |
| | **DSR** | $0.3365_{(\pm0.029)}$ | $14.94_{(\pm0.49)}$ | $1746_{(\pm1.8e+02)}$ | $1.343_{(\pm0.1)}$ |
| | **EPLEX** | $0.7375_{(\pm0.027)}$ | $52.64_{(\pm1.1)}$ | $11450_{(\pm690)}$ | $3.355_{(\pm0.15)}$ |
| | **GP-GOMEA** | $0.8808_{(\pm0.02)}$ | $34.77_{(\pm0.97)}$ | $4786_{(\pm4.4e+02)}$ | $2.537_{(\pm0.14)}$ |
| | **GPlearn** | $0.468_{(\pm0.03)}$ | $67.71_{(\pm16)}$ | $3606_{(\pm3.6e+02)}$ | $2.126_{(\pm0.15)}$ |
| | **Operon** | $0.9392_{(\pm0.013)}$ | $68.73_{(\pm1.4)}$ | $1947_{(\pm64)}$ | $3.162_{(\pm0.11)}$ |
| | **PySR** | $0.6762_{(\pm0.064)}$ | $9.219_{(\pm0.46)}$ | $791.6_{(\pm44)}$ | $2.114_{(\pm0.2)}$ |
| | **RSRM** | $0.2455_{(\pm0.025)}$ | $13.4_{(\pm0.37)}$ | $116.9_{(\pm3)}$ | $1.435_{(\pm0.092)}$ |
| | **SBP-GP** | $0.9365_{(\pm0.015)}$ | $513.4_{(\pm12)}$ | $2.768e+04_{(\pm2.4e+02)}$ | $7.872_{(\pm0.34)}$ |
| | **SR4MDL** | $0.6271_{(\pm0.026)}$ | $21.7_{(\pm0.79)}$ | $470_{(\pm34)}$ | $1.58_{(\pm0.086)}$ |
| | **GP** | $0.7215_{(\pm0.029)}$ | $34.61_{(\pm1.7)}$ | $1728_{(\pm67)}$ | $2.367_{(\pm0.12)}$ |
| | **EIC-GP** | $0.7288_{(\pm0.038)}$ | $32.47_{(\pm2.2)}$ | $1718_{(\pm90)}$ | $2.136_{(\pm0.15)}$ |
| | $\Delta(\%)$ | 1% | -7% | -0.50% | -9.80% |
| | **MCTS** | $0.6917_{(\pm0.046)}$ | $17.68_{(\pm1.2)}$ | $10580_{(\pm1000)}$ | $1.185_{(\pm0.14)}$ |
| | **EIC-MCTS** | $0.7107_{(\pm0.045)}$ | $17.64_{(\pm1.2)}$ | $12080_{(\pm1.1e+03)}$ | $0.9983_{(\pm0.12)}$ |
| | $\Delta(\%)$ | 2.70% | -0.20% | 14% | -16% |

### C.2  Enhance Generative Methods with EIC

We demonstrate that EIC improves the alignment between pre-training formulas and real-world physical formulas by measuring the similarity between randomly generated formulas and EIC-filtered formulas with three datasets of real physical formulas. These datasets include the Feynman dataset, the Strogatz ODE dataset, and the Wiki Named Equation dataset. The first two are derived from the white-box models in SRBench, while the third was collected by Guimerà et al. (2020) from Wikipedia, containing over a thousand named formulas. We cleaned and deduplicated the formulas, removing those with special operators (e.g., matrix operations) and inherently redundant formulas

Table 4: Whitebox Results at 0.001 noise

| Type | algorithm | $R^2 > 0.99$ | complexity | duration | EIC |
|---|---|---|---|---|---|
| Regression | FEAT | 0.6236 (±0.03) | 186.4 (±7.2) | 1598 (±89) | 3.556 (±0.18) |
| Generative | E2ESR | 0.2595 (±0.03) | 89.56 (±2.1) | 3.846 (±0.13) | 3.641 (±0.07) |
| | NeurSR | 0.0767 (±0.01) | 31.34 (±0.26) | 23.52 (±0.39) | 3.309 (±0.05) |
| | SNIP | 0.1579 (±0.02) | 25.2 (±0.39) | 1.865 (±0.07) | 2.759 (±0.11) |
| Search | AFP | 0.5646 (±0.03) | 39.27 (±1) | 3327 (±104) | 2.954 (±0.14) |
| | AFP-FE | 0.7308 (±0.02) | 46.71 (±1.1) | 24216 (±511) | 3.252 (±0.17) |
| | AIFeynman2 | 0.8305 (±0.02) | 118.7 (±20) | 575.1 (±27) | 1.974 (±0.25) |
| | BSR | 0.2515 (±0.02) | 27.12 (±0.56) | 29613 (±1691) | 3.709 (±0.1) |
| | DSR | 0.3815 (±0.03) | 16.32 (±0.47) | 794.6 (±27) | 1.273 (±0.09) |
| | EPLEX | 0.77 (±0.02) | 55.3 (±0.8) | 11057 (±318) | 3.66 (±0.14) |
| | GP-GOMEA | 0.9038 (±0.02) | 44.79 (±0.77) | 2678 (±197) | 3.811 (±0.13) |
| | GPlearn | 0.4809 (±0.03) | 57.12 (±8.4) | 3054 (±234) | 2.093 (±0.13) |
| | Operon | 0.9538 (±0.01) | 69.39 (±1.4) | 1967 (±70) | 3.276 (±0.10) |
| | PySR | 0.7914 (±0.07) | 9.381 (±0.5) | 658.6 (±34) | 2.029 (±0.23) |
| | RSRM | 0.263 (±0.03) | 13.22 (±0.47) | 128.3 (±2.5) | 1.367 (±0.12) |
| | SBP-GP | 0.9362 (±0.01) | 600.9 (±9.4) | 27934 (±172) | 8.697 (±0.32) |
| | SR4MDL | 0.6099 (±0.03) | 26.8 (±0.65) | 540.5 (±28) | 1.855 (±0.08) |
| | GP | 0.691 (±0.05) | 35.65 (±2.3) | 1813 (±103) | 2.811 (±0.16) |
| | EIC-GP | 0.7103 (±0.04) | 31.7 (±2.2) | 1703 (±103) | 2.394 (±0.16) |
| | Δ(%) | +3% | -11% | -6.08% | -14.82% |
| | MCTS | 0.6404 (±0.03) | 19.3 (±0.6) | 3641 (±469) | 1.246 (±0.09) |
| | EIC-MCTS | 0.6692 (±0.08) | 20.46 (±1.8) | 15228 (±2045) | 0.9275 (±0.18) |
| | Δ(%) | +4.49% | +6.03% | +318% | -26% |

Table 5: Whitebox Results at 0.01 noise

| Type | algorithm | $R^2 > 0.99$ | complexity | duration | EIC |
|---|---|---|---|---|---|
| Regression | FEAT | 0.6198 (±0.03) | 159.3 (±6.2) | 1357 (±73) | 3.464 (±0.2) |
| Generative | E2ESR | 0.2388 (±0.03) | 97.61 (±2.3) | 4.175 (±0.1) | 4.015 (±0.08) |
| | NeurSR | 0.0759 (±0.01) | 31.44 (±0.25) | 23.66 (±0.38) | 3.292 (±0.04) |
| | SNIP | 0.1414 (±0.02) | 27.05 (±0.37) | 2.109 (±0.08) | 3.407 (±0.11) |
| Search | AFP | 0.5808 (±0.03) | 40.62 (±1) | 3666 (±118) | 3.152 (±0.16) |
| | AFP-FE | 0.7262 (±0.02) | 47.12 (±1.1) | 25731 (±414) | 3.413 (±0.17) |
| | AIFeynman2 | 0.797 (±0.02) | 140.2 (±23) | 562.8 (±28) | 2.276 (±0.27) |
| | BSR | 0.2685 (±0.02) | 29.19 (±0.71) | 29680 (±1554) | 3.961 (±0.12) |
| | DSR | 0.3838 (±0.03) | 16.45 (±0.48) | 882.8 (±33) | 1.289 (±0.09) |
| | EPLEX | 0.7792 (±0.02) | 53.9 (±0.82) | 9901 (±240) | 3.711 (±0.15) |
| | GP-GOMEA | 0.9085 (±0.02) | 44.46 (±0.72) | 2777 (±197) | 3.919 (±0.13) |
| | GPlearn | 0.4813 (±0.03) | 56.85 (±11) | 3084 (±235) | 2.064 (±0.13) |
| | Operon | 0.9438 (±0.01) | 87.29 (±0.42) | 2835 (±80) | 4.908 (±0.1) |
| | PySR | 0.7857 (±0.07) | 9.545 (±0.5) | 803.6 (±22) | 2.034 (±0.22) |
| | RSRM | 0.2703 (±0.03) | 13.38 (±0.37) | 132.7 (±2.3) | 1.384 (±0.09) |
| | SBP-GP | 0.9338 (±0.01) | 623 (±9.2) | 28074 (±154) | 8.804 (±0.31) |
| | SR4MDL | 0.6084 (±0.03) | 26.6 (±0.64) | 805.6 (±41) | 1.876 (±0.08) |
| | GP | 0.6642 (±0.05) | 34.83 (±2) | 1784 (±100) | 2.684 (±0.16) |
| | EIC-GP | 0.6884 (±0.05) | 33.65 (±2.2) | 1582 (±104) | 2.554 (±0.16) |
| | Δ(%) | +4% | -3% | -11.31% | -4.84% |
| | MCTS | 0.6186 (±0.03) | 19.33 (±0.63) | 4158 (±464) | 1.226 (±0.09) |
| | EIC-MCTS | 0.7083 (±0.08) | 18.99 (±1.8) | 15996 (±1928) | 0.9407 (±0.2) |
| | Δ(%) | +14.51% | -1.77% | +285% | -23% |

Table 6: Whitebox Results at 0.1 noise

| Type | algorithm | $R^2 > 0.99$ | complexity | duration | EIC |
|---|---|---|---|---|---|
| Regression | **FEAT** | 0.4606 (±0.03) | 97.84 (±3.8) | 742.3 (±32) | 2.891 (±0.17) |
| Generative | **E2ESR** | 0.0176 (±0.01) | 103.3 (±2.1) | 6.119 (±0.34) | 4.841 (±0.1) |
| | **NeurSR** | 0.0421 (±0.01) | 31.83 (±0.24) | 24.93 (±0.39) | 3.344 (±0.04) |
| | **SNIP** | 0.0346 (±0.01) | 30.95 (±0.35) | 2.655 (±0.1) | 4.862 (±0.11) |
| Search | **AFP** | 0.5485 (±0.03) | 41.18 (±0.99) | 3485 (±95) | 3.535 (±0.17) |
| | **AFP-FE** | 0.7262 (±0.02) | 49.1 (±1) | 26795 (±319) | 4.077 (±0.19) |
| | **AIFeynman2** | 0.1949 (±0.02) | 157.3 (±21) | 629.5 (±91) | 3.695 (±0.26) |
| | **BSR** | 0.22 (±0.02) | 31.01 (±0.84) | 31506 (±2521) | 4.104 (±0.11) |
| | **DSR** | 0.3823 (±0.03) | 16.3 (±0.47) | 781 (±24) | 1.288 (±0.09) |
| | **EPLEX** | 0.7585 (±0.02) | 46.46 (±0.95) | 9219 (±192) | 3.394 (±0.15) |
| | **GP-GOMEA** | 0.8885 (±0.02) | 46.13 (±0.7) | 2886 (±207) | 4.258 (±0.12) |
| | **GPlearn** | 0.4786 (±0.03) | 46.32 (±6.3) | 2715 (±209) | 1.952 (±0.13) |
| | **Operon** | 0.8892 (±0.02) | 88.61 (±0.32) | 2768 (±71) | 5.29 (±0.1) |
| | **PySR** | 0.6234 (±0.08) | 10.49 (±0.78) | 798 (±17) | 1.797 (±0.23) |
| | **RSRM** | 0.2543 (±0.03) | 13.11 (±0.34) | 134.2 (±2.1) | 1.379 (±0.09) |
| | **SBP-GP** | 0.8662 (±0.02) | 652 (±8.9) | 28173 (±140) | 9.227 (±0.31) |
| | **SR4MDL** | 0.4717 (±0.03) | 29.81 (±0.59) | 833.9 (±44) | 2.577 (±0.07) |
| | **GP** | 0.3258 (±0.05) | 29.67 (±1.7) | 1157 (±96) | 2.938 (±0.15) |
| | **EIC-GP** | 0.3182 (±0.05) | 29.97 (±1.9) | 861.7 (±91) | 2.823 (±0.14) |
| | $\Delta(\%)$ | -2% | +1% | -25.51% | -3.92% |
| | **MCTS** | 0.5064 (±0.06) | 18.77 (±1.1) | 2778 (±325) | 1.174 (±0.17) |
| | **EIC-MCTS** | 0.5012 (±0.06) | 18.67 (±1.3) | 3005 (±403) | 1.092 (±0.16) |
| | $\Delta(\%)$ | -1% | +0.5% | +8.17% | -6.98% |

Table 7: blackbox results

| Type | algorithm | $R^2$ | complexity | EIC |
|---|---|---|---|---|
| Regression | **FEAT** | 0.7621 (±0.01) | 82.49 (±3.3) | 1.441 (±0.09) |
| Generative | **E2ESR** | 0.3612 (±0.02) | 61.09 (±1) | 3.581 (±0.13) |
| | **NeurSR** | 0.1228 (±0.01) | 13.33 (±0.12) | 2.208 (±0.07) |
| | **SNIP** | 0.3335 (±0.02) | 38.91 (±0.55) | 3.307 (±0.14) |
| Search | **AFP** | 0.6333 (±0.01) | 34.89 (±1) | 2.941 (±0.12) |
| | **AFP-FE** | 0.64 (±0.01) | 36.04 (±1) | 3.091 (±0.12) |
| | **AIFeynman2** | 0.211 (±0.02) | 2240 (±250) | 3.248 (±0.14) |
| | **BSR** | 0.2725 (±0.02) | 22.52 (±0.91) | 2.92 (±0.08) |
| | **DSR** | 0.5625 (±0.01) | 9.465 (±0.26) | 2.408 (±0.07) |
| | **EPLEX** | 0.7372 (±0.01) | 53.14 (±0.73) | 3.315 (±0.11) |
| | **GP-GOMEA** | 0.7381 (±0.01) | 30.27 (±0.96) | 2.996 (±0.08) |
| | **GPlearn** | 0.539 (±0.01) | 19.06 (±0.96) | 2.151 (±0.08) |
| | **Operon** | 0.7945 (±0.02) | 65.69 (±1.3) | 3.264 (±0.06) |
| | **SBP-GP** | 0.7869 (±0.01) | 634 (±18) | 6.252 (±0.22) |
| | **SR4MDL** | 0.6258 (±0.01) | 29.88 (±0.56) | 2.267 (±0.10) |
| | **GP** | 0.6881 (±0.04) | 34.84 (±2.4) | 2.452 (±0.18) |
| | **EIC-GP** | 0.6433 (±0.05) | 27.48 (±2.1) | 2.202 (±0.21) |
| | $\Delta(\%)$ | -6.50% | -26.00% | -10% |
| | **MCTS** | 0.6038 (±0.06) | 35.91 (±1.9) | 2.463 (±0.27) |
| | **EIC-MCTS** | 0.6227 (±0.05) | 33.89 (±1.9) | 1.951 (±0.2) |
| | $\Delta(\%)$ | +3.13% | -5.62% | -20.80% |

(e.g., numerous occurrences of simple expressions like $a \times b$, retaining only one). The final set contains 940 physical formulas.

In addition to the training process we reported in Figure 7, we also provide the training process of SNIP and SR4MDL in Figure 9 and Figure 10. The SNIP model trained on random formulas reaches a final $R^2$ performance of $0.5299$, while it trained on EIC-filtered formulas reaches a final $R^2$ of $0.6016$, which is $13.5\%$ higher. For the SR4MDL, trained on random formulas, it reaches final RMSE and MAE of $8.6778$ and $6.9254$, respectively. While trained on EIC-filtered formulas, it can reach final RMSE and MAE of $8.2319$ and $6.5763$, which are $5.14\%$ and $5.04\%$ higher, respectively.

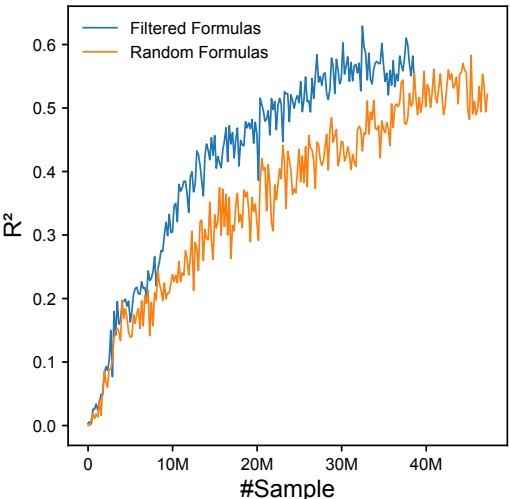

Figure 9: Training process of SNIP using both random and filtered formulas

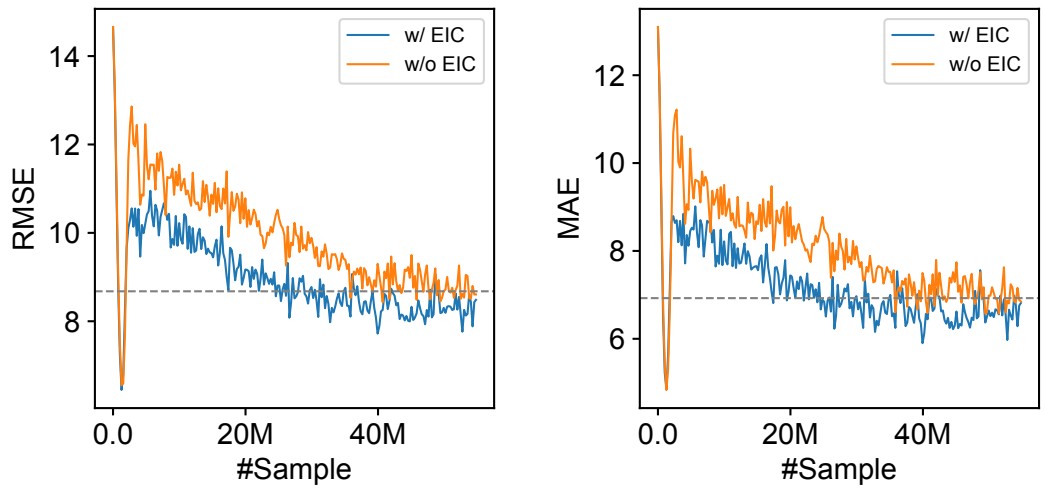

Figure 10: Training process of SR4MDL using both random and filtered formulas

### C.3 EXPERT EVALUATION RESULTS

To demonstrate that EIC evaluates formula interpretability in a manner consistent with human intuition, we compared formulas discovered by EIC-MCTS and PySR. These two methods were chosen because, as shown in Section 4.2, they achieved the lowest average EIC scores (EIC-MCTS) and

the lowest average complexity (PySR), respectively, while exhibiting comparable $R^2$ performance. We focused on the Pareto fronts obtained from 19 one- or two-dimensional black-box problems in SRBench, as these problems are easier for experts to interpret through visual inspection. From each pair of Pareto fronts, we selected formula pairs by minimizing the distance function:

$$L(f_1, f_2) = (\text{Complexity}[f_1] - \text{Complexity}[f_2])^2 + (R^2[f_1] - R^2[f_2])^2 - (\text{EIC}[f_1] - \text{EIC}[f_2])^2, \quad (8)$$

subject to the following constraints:

$$\begin{aligned}
|\text{Complexity}[f_1] - \text{Complexity}[f_2]| \leq 2, \qquad |R^2[f_1] - R^2[f_2]| \leq 0.02, \\
|\text{EIC}[f_1] - \text{EIC}[f_2]| \geq 3, \qquad\qquad \max(R^2[f_1], R^2[f_2]) > 0.85,
\end{aligned} \quad (9)$$

where $f_1$ and $f_2$ traverse each formula on the Pareto front of PySR and EIC-MCTS, respectively. This procedure ensured that selected pairs exhibited similar complexity and accuracy but substantially different EIC values. In total, 172 formula pairs were obtained across the 19 Pareto fronts.

Each pair was then presented to large language models (LLMs) acting as a domain expert to select the preferred one. To provide independent evaluations of interpretability, we employed two large language models: GPT-4o-Mini and Qwen3, which are trained on different linguistic distributions and can thus reduce the risk of model-specific bias. We format the prompt based on the format shown in Figure 13, where we provided the LLMs with the dataset name, textual description, $(x, y)$ sampled points, the mathematical expressions of the two formulas, as well as their $R^2$ values and complexities. Importantly, the EIC scores were withheld from the models to ensure unbiased evaluation. We evaluate every formula pair on every LLM five times under a temperature setting of 0.7 to account for sampling variability. The result is shown in Figure 14, where LLM shows a preference of 72.19% to formulas discovered by EIC, which is similar to that of the human experts, demonstrating the reliability of the results.

For the human rating experiment, we invite 108 volunteer participants with at least a bachelor's degree in science or engineering. Each participant was randomly assigned 10 formula pairs. For each pair, participants received the data samples visualization, dataset description, and the two candidate formulas with their $R^2$ and complexity, and were asked to choose the more interpretable one (EIC scores were withheld), as demonstrated in Figure 11 and 12. We collected 1080 evaluations in total. After discarding responses completed in under 60 seconds, 840 evaluations remained, with each pair assessed on average 4.9 times.

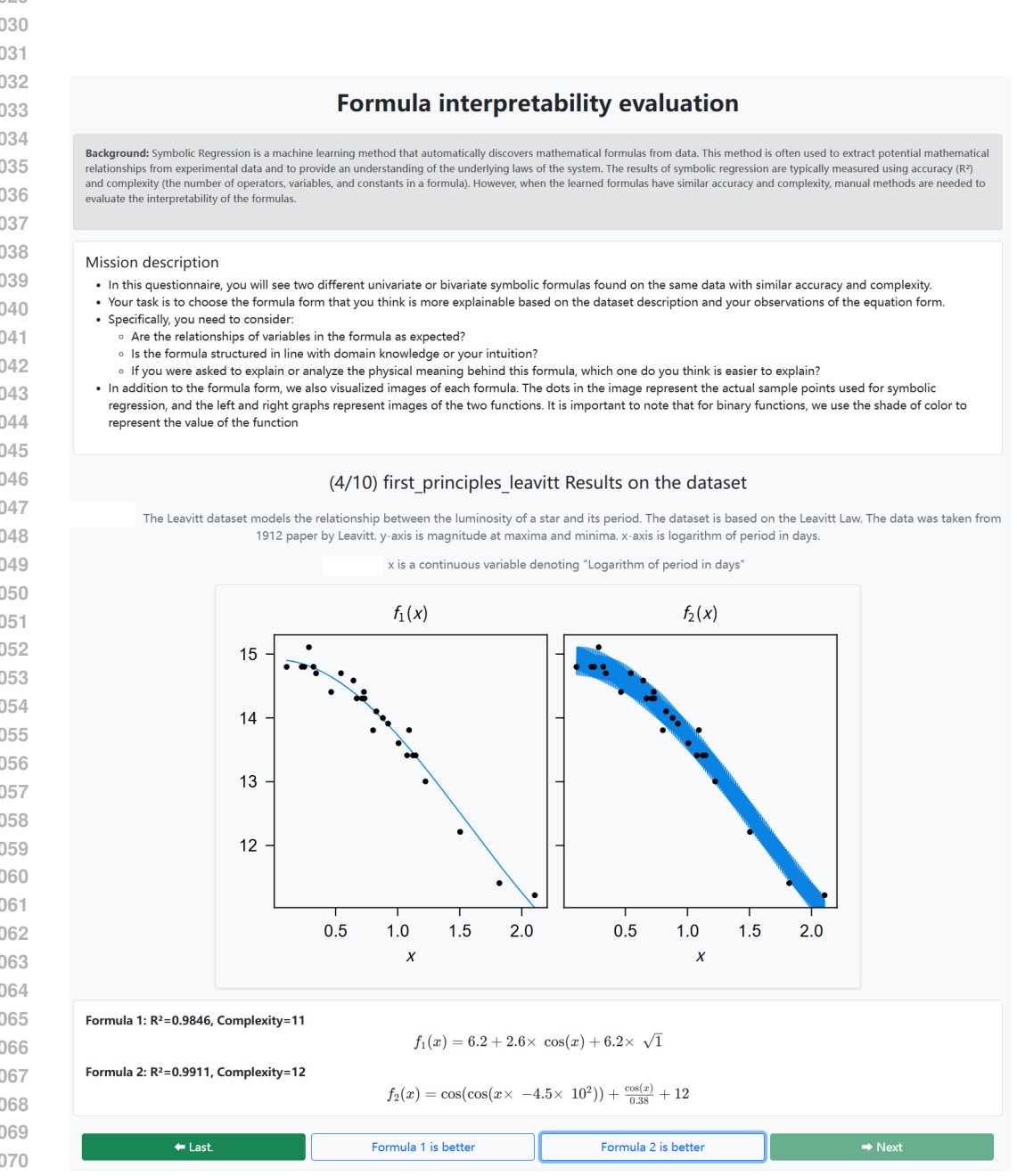

Figure 11: Website interface used by human experts for scoring, where a one-dimensional dataset is demonstrated

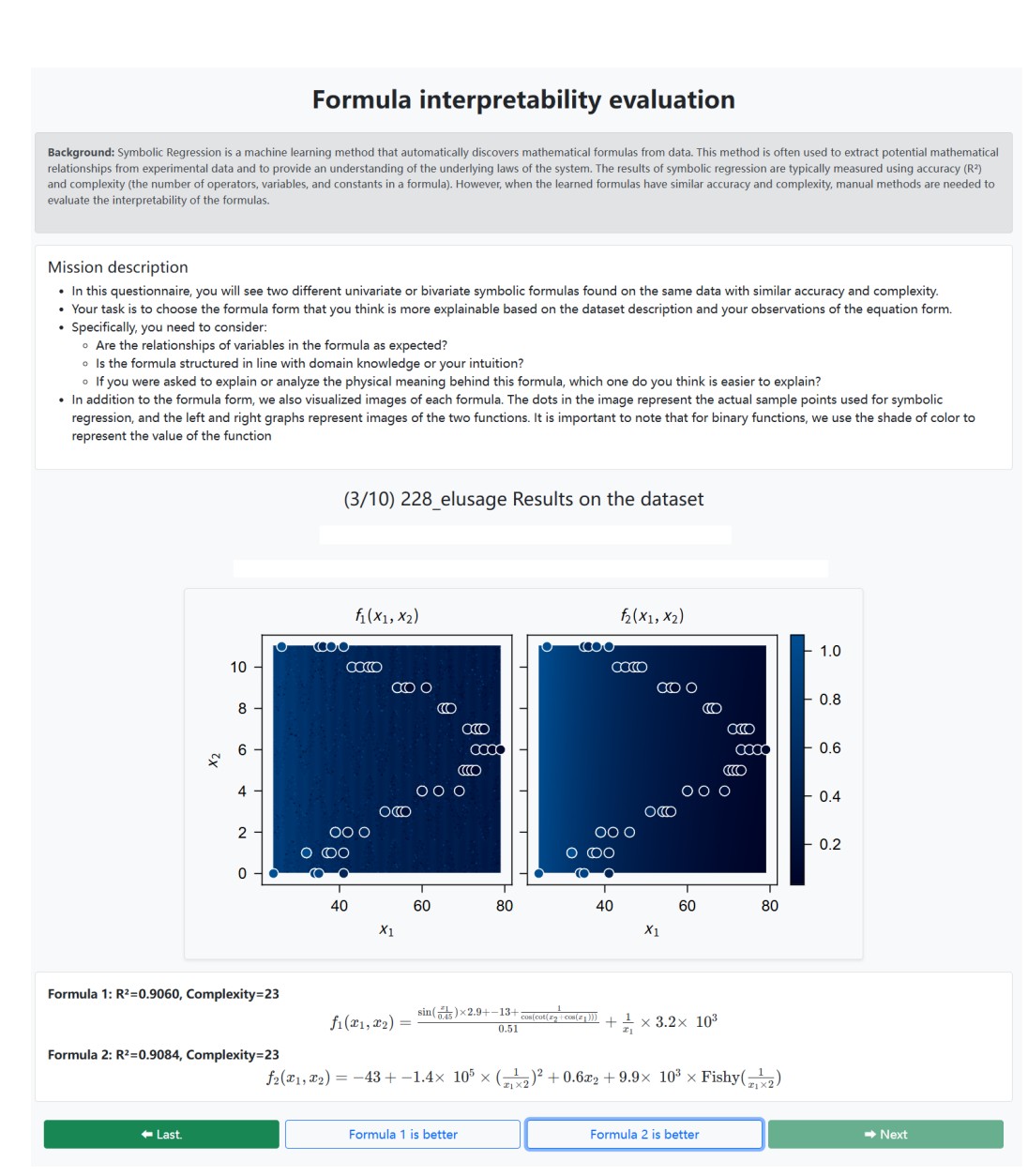

Figure 12: Website interface used by human experts for scoring, where a two-dimensional dataset is demonstrated

```
Assume you're an expert. Your have derived two expressions describing a absorption dataset you're
studying using symbolic regression. They have similar R² and formula lengths, making them
indistinguishable by accuracy or complexity. Therefore, you need to analyze the two equations based
on your domain knowledge, and choose a formula you prefer.

- **Dataset Description**: A real-world dataset containing the absorption of light for a solution
containing a specific molecule at different levels of concentration. The original publication has
data for 4 different molecules, and here we include data from only one of them (`real_data/
absorption/examples/example0.csv`), the one with highest number of samples.

- **Variable Description**: a continuous variable `Xaxis0` denoting the Concentration (mol/L).
- **Data Samples (x, y)**: [(0.2,0.062),(0.5,0.22),(1,0.25),(1.5,0.38),(2.3,0.53),(3,0.62),(4,0.85),
(5,1),(6,1.2),(7.2,1.5),(10,2),(15,2.8),(30,3.3),(45,3.4)]
- **Result 1**:
    - R2: 0.9541
    - Formula Length: 5
    - Equation: log(Xaxis0 - log(Xaxis0))
- **Result 2**:
    - R2: 0.9436
    - Formula Length: 6
    - Equation: -0.265 + 0.623 * sqrt(Xaxis0)

Your response should end with a single line indicating your choice: either \"I prefer Equation 1\"
or "I prefer Equation 2".
```

Figure 13: Prompt we used to ask LLM to act as domain experts to evaluate formula interpretability.

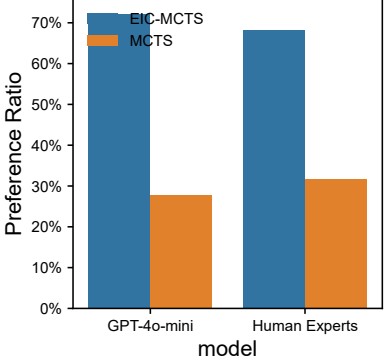

Figure 14: Results of LLM and human experts' preferences for formula interpretability.

