# OpenReview forum: "Beyond Formula Complexity: Effective Information Criterion Improves Performance and Interpretability for Symbolic Regression"
_ICLR.cc/2026/Conference — Submitted to ICLR 2026_

### Official Review · Reviewer_RqEj · 2025-10-21

**Soundness:** 3
**Presentation:** 2
**Contribution:** 2
**Rating:** 4
**Confidence:** 3

**Summary:**

This paper proposes a new evaluation metric in the field of symbolic regression—the Effective Information Criterion (EIC). This metric relates "the loss of significant digits or the amplification of rounding noise as data flows through the system" to the structural reasonableness of formulas, assessing expression quality from the perspective of numerical stability. By incorporating EIC into the evaluation and search process to filter out numerically ill-conditioned expressions, the authors claim that this criterion helps generate more interpretable formulas.

**Strengths:**

1) EIC introduces a new evaluation metric from the perspectives of numerical analysis and information theory, to some extent addressing the limitations of traditional complexity and accuracy measures.
2) EIC is able to filter some unreasonable nested expressions (e.g., exp(exp(exp(x)))) or numerically ill-conditioned formulas.
3) The authors validate EIC within GP and MCTS-based symbolic regression frameworks, across multiple scenarios including noisy data, black-box problems, and human interpretability assessments.
4) On the SRBench dataset, GP methods incorporating EIC achieve complexity–accuracy Pareto frontier performance.

**Weaknesses:**

(1) Lack of in-depth analysis and validation regarding the effect and generalization ability of EIC across different datasets.
(2) Absence of intuitive evidence demonstrating the practical improvement and interpretability brought about by the EIC.
(3) Insufficient discussion regarding the potential limitations of EIC
(4) Some experimental details are not clear.

**Questions:**

1)	On the white-box subset of SRBench, is the improvement consistent across most datasets, or concentrated in specific ones? What proportion of datasets show improvement, and what is the average improvement magnitude? Are there notable cases of accuracy degradation? Additionally, have the authors analyzed characteristics of datasets that benefit most (e.g., variable count, input scaling) to assess the generality of the method?
2)	In scientific domains where variables exhibit drastically different numerical scales, does EIC’s evaluation criterion affect formula discovery under real sampling ranges? For example, in the SRSD dataset, where variables and constants are physically meaningful, some formulas involve variables with ranges $U_{log}(10^{-29},10^{-27}) and U_{log}(10^6,10^8)$ . How does EIC behave under such multi-scale conditions?
3)	Since the core contribution lies in proposing EIC, it would be informative to show representative formula examples with and without EIC (e.g., EIC-GP vs. vanilla GP). Listing several illustrative formulas would make it easier to see how EIC improves interpretability in practice.
4)	The paper claims that EIC facilitates discovery of true underlying formulas. Could the authors report the solution rate improvement? Moreover, as $R^2 > 0.999$ typically indicates precise fits, it would be helpful to report how many formulas reach this stricter threshold under the EIC-based approach.
5)	Please clarify whether the reported $R^2$ values are based on the median or the mean across runs.
6)	In the "expert preference experiment" (Section 4.4), it is recommended to report inter-rater agreement (e.g., Fleiss’ κ) to strengthen the statistical reliability of expert evaluations.
7)	Since EIC measures the amplification of propagated numerical errors, would directly applying it during the search process risk false penalization of inherently sensitive but valid functions (e.g., $exp(x^2 + y^2)$ or Gaussian functions)?
8)	Please check the paper for formatting and typographical issues. For example, line 456 should refer to Figure 8 (not Figure 14), and “Implementaion” should be corrected to “Implementation”.

---

### Official Review · Reviewer_6Kve · 2025-10-29

**Soundness:** 1
**Presentation:** 3
**Contribution:** 2
**Rating:** 2
**Confidence:** 5

**Summary:**

The paper proposes the “Effective Information Criterion” (EIC), which is intended to have physical meaning (i.e., significant digits loss, noise gain) for equations. EIC is then used as a proxy for interpretability in symbolic regression (SR), which is argued to be better than length in SR. The paper then uses EIC as a form of regularization in SR algorithms, and they also perform experiments to check the alignment of EIC with what human experts think is interpretable.

**Strengths:**

The presentation, details and motivation from lines 1 to 188 are of high quality. The paper introduces the work in an intuitive manner that makes it seem natural that this is a ‘right’ approach.

I am convinced by the paper that the motivation of EIC is well justified and would be impactful if executed well.

The alignment with human experiments is valuable experiment to the SR community and is a strong addition to their other results.

**Weaknesses:**

Multiple errors at the core of the paper, specifically from line 189 onwards, with some errors which I am not even sure are fixable. I list the key reasons which affected my score below. **If addressed adequately in the rebuttal, I will increase the score by multiple stages to be in the acceptance range**.

First, in line 189, the equation given, f(x) = (x+10^100) – 10^100, is given as an example with suffers from “catastrophic cancellation”, is this not just a data structure/programming issue? For example, is this not just an issue of using the wrong data structure for numerical precision? Also, simplification (which has been done in SR [1]) mechanisms can remove this error. It is not clear why this example is relevant to EIC, is the EIC of this equation not just 0, according to equation (4)? What is the point of this equation when on one hand the paper claims it is “bad” because it suffers from catastrophic cancellation, yet under the new contribution, EIC, it is perfect at a value of 0? Is this not a contradiction?

[1] Cao, Lulu, et al. "Genetic programming symbolic regression with simplification-pruning operator for solving differential equations."

Second, it seems that the computation in the experiments highly depend on the underlying data structure (e.g., float64, binary128, etc., or even a custom one) used to perform computations. This is not stated clearly in the paper, and I am unable to find what computation precision is used for the experiments.

Third, the LARGEST error, which is the key reason for the "reject” score, is in lines 207-208, when giving the equation “$M = 1 - \frac{1}{2} \log_{10}(12\delta_r^2)$”. This relation is a major part of EIC, and affects the correctness of the WHOLE paper. There is no justification for how the equation is obtained, and it is flawed if the paper is assuming that the output noise is uniform. Is the paper saying that for equations in SR, say f(x)=1/x, injecting a uniform random noise to x will result in a uniform noise to f(x) as well? I suspect the paper made the serious mistake of assuming that uniform input noise will result in uniform output noise, which is not true for many equations in SR (e.g., non-linear operators), even with small noise.

Fourth, lines 194 to 196 is problematic too. The paper claims “truncating a variable x to N significant digits can be expressed as…”. This can be disproved with a simple counter example, where x=3, and we choose N=1, then using the definitions (e.g., the range of noise in line 195 and M=log_10|x| in line 196), x_hat ranges from about 1.5 to 4.5. Here, 1.5 is not a truncation of x=3 to N=1 significant digits and 4.5 is not a truncation of x=3 to N=1 significant digits either. This affects the “physical meaning” of EIC.

Fifth, the white-box and black-box results are inconsistent in the paper itself. For instance, there are algorithms which are present in Figure 6 which are NOT present in the corresponding Table 7. Likewise, there are algorithms which are present in Table 7 which are NOT present in the corresponding Figure 6. Also, using aggregated “rank” in Figures 5 and 6 (note: the computation of ranking is not clear in the paper so I am assuming the same process as SRBench is used) has been shown to lead to paradox where the algorithms on the Pareto frontier is different when the absolute value of the metrics are used instead of “ranks” [2]. It would be better to show a plot with the axis using the absolute value of the metrics, instead of aggregated ranks, this way, the absolute difference in metric values can be seen as well. Finally, confidence intervals/error bars should be added to the plot as well, as done in the EIC plot.

[2] Fong, Kei Sen, and Mehul Motani. "Pareto-Optimal Fronts for Benchmarking Symbolic Regression Algorithms."

**Questions:**

Please address each of the 5 weaknesses above, especially the third weakness. Critically, for the third weakness, please provide a clear, step by step, reasoning for how “$M = 1 - \frac{1}{2} \log_{10}(12\delta_r^2)$” is obtained.

---

> ### Author Response · Authors · 2025-11-28
> **Response to Reviewer 6Kve**
>
> We thank the reviewer for the detailed feedback and for recognizing the high quality of our motivation and presentation, as well as the value of our human alignment experiments.
>
> However, we respectfully submit that the concerns regarding the catastrophic cancellation example and the derivation of rounding noise stem from **factual misunderstandings**. As demonstrated below, a formula exhibiting catastrophic cancellation and its algebraically equivalent simplified form correspond to distinct computational processes and thus yield different EIC scores, reflecting EIC's capability to identify structural anomalies. Furthermore, according to Widrow's Quantization Theorem, the complex non-linear distributions of variables and outputs are fully compatible with the uniform distribution of their rounding noise. We hope the clarifications below can fully resolve these misunderstandings and justify a re-evaluation of our work.

---

> > ### Author Response · Authors · 2025-12-03
> > **Catastrophic Cancellation and EIC Value**
> >
> > **Reviewer Concern**: *"Is this formula $(x+10^{100})−10^{100}$ not just a data structure issue? Is the EIC not just 0?"*
> >
> > **Response**: The example $(x+10^{100}) - 10^{100}$ is not merely a programming or data structure issue, but a classic example of **catastrophic cancellation**. As noted in the Wikipedia entry for catastrophic cancellation (it cites an analogous example of $x-y$ where $x=1+10^{-29}$ and $y=1+10^{-30}$, which is essentially the same as the example here.), this phenomenon demonstrates precision loss caused by an unreasonable **computational process** inherent to the formula's structure, independent of the computing medium (e.g., chosen precision). While one could bypass this error by using arbitrarily high-precision arithmetic, the inherent precision loss reveals an anomalous structure within the formula, implying unreasonableness and poor interpretability.
> >
> > Consider $f_1(x)=x$ and $f_2(x)=(x+10^{100})-10^{100}$. Although they are algebraically equivalent, they reflect distinct computational processes. The latter requires retaining over 100 significant digits during the evaluation of the first bracket to avoid losing all information about $x$. Consequently, our method yields $EIC(f_1) \approx 0$, whereas $EIC(f_2) \approx 100$. This example demonstrates how a higher EIC score correctly identifies anomalous formula forms (or more precisely, the unstable computational processes they represent).

---

> > > ### Author Response · Authors · 2025-12-03
> > > **Underlying Data Structure**
> > >
> > > **Reviewer Concern**: *"Computations depend on underlying data structure... I am unable to find what computation precision is used."*
> > >
> > > **Response**: We clarify that all experiments were conducted using the standard IEEE 754 64-bit floating-point format (float64 in Python). However, we emphasize that **our results are robust to this choice due to the asymptotic invariance of EIC**. Specifically, once the injected rounding noise strength $\sigma$ is small enough, EIC will not change as $\sigma$ further decreases. This can be demonstrated by the following code. As shown in the output, EIC stabilizes rapidly once $\sigma$ (`noise_level`) drops below $0.1$. Consequently, fixing $\sigma=10^{−6}$ (as done in the paper) captures the stable, asymptotic behavior of the formula structure, making the specific choice of $\sigma$ irrelevant and making the EIC (asymptotically) parameter-free.
> > >
> > > ```python
> > > import numpy as np
> > > import nd2py as nd
> > > from src.metrics.eic import get_eic
> > > l0 = Variable('l0')
> > > v = Variable('v')
> > > X = {
> > >     'l0': np.random.uniform(1, 10, 1000),
> > >     'v': np.random.uniform(1e5, 1e7, 1000)
> > > }
> > > l = l0 * sqrt(1 - (v / 299792458)**2)
> > > print(get_eic(l, X, random_state=42, noise_level=1)) # 0.5832177066263028
> > > print(get_eic(l, X, random_state=42, noise_level=0.1)) # 0.3431258579326261
> > > print(get_eic(l, X, random_state=42, noise_level=0.01)) # 0.3448326523799693
> > > print(get_eic(l, X, random_state=42, noise_level=0.001)) # 0.3453428768175242
> > > print(get_eic(l, X, random_state=42, noise_level=0.0001)) # 0.3453972675850197
> > > print(get_eic(l, X, random_state=42, noise_level=0.00001)) # 0.34540274030306733
> > > print(get_eic(l, X, random_state=42, noise_level=0.000001)) # 0.3454032879102719
> > > ```
> > >
> > > Therefore, using float128 or arbitrary-precision arithmetic would yield the same relative EIC rankings. The use of float64 is a practical choice that does not compromise the theoretical validity of the metric.

---

> > > > ### Author Response · Authors · 2025-12-03
> > > > **Derivation of the Error Term**
> > > >
> > > > **Reviewer Concern**: *"Assuming that uniform input noise will result in uniform output noise... which is not true for non-linear operators."*
> > > >
> > > > **Response**: We believe there is a **misunderstanding** regarding the noise source. We do not assume that the input noise propagates to become uniform output noise. Instead, the uniform distribution assumption applies strictly to the rounding (or quantization) error introduced at the output step, based on Widrow's Quantization Theorem. Specifically, this involves two types of noise: the **Gaussian noise at the input** (which can be transformed into an arbitrarily complex distribution after the nonlinear operator in the formula is applied), and the **rounding noise** caused by maintaining finite precision during computation, which, according to the Widrow's Quantization Theorem, **strictly tends to a uniform distribution as long as the quantization precision is sufficiently high**. That is, as long as the number of digits kept $N$ is large enough (or equivalent, $\sigma$ is small enough), the rounding error will converge to a uniform distribution $U(-\Delta/2, \Delta/2)$, where $\Delta = 10^{-N}$ is the quantization interval, regardless of the distribution of $f(x)$.

---

> > > > > ### Author Response · Authors · 2025-12-03
> > > > > **Truncation Noise Formulation**
> > > > >
> > > > > **Reviewer Concern:** *"The claim about truncating $x$ to $N$ significant digits... can be disproved with a simple counter example, where $x=3$..."*
> > > > >
> > > > > **Response:** We clarify that while truncating a single integer point (like $x=3$) is a deterministic operation that may yield zero error, **Symbolic Regression deals with datasets where $x$ follows a distribution**, rather than a single scalar value. In this statistical context, truncating every data point $x_i$ in a dataset to $N$ significant digits effectively introduces a rounding error to each $x_i$. As established in Widrow's Quantization Theorem (discussed in the response above), when $x$ varies across a continuous domain, these rounding errors collectively converge to the Uniform distribution described in our paper.

---

### Official Review · Reviewer_UdbF · 2025-11-04

**Soundness:** 3
**Presentation:** 2
**Contribution:** 1
**Rating:** 4
**Confidence:** 5

**Summary:**

This paper proposes a new criterion, EIC (Effective Information Criterion), which extends the traditional concept of formula complexity to encourage more rational structure discovery in symbolic regression and scientific modeling.
EIC balances data fit and structural information and is presented as a generalization of AIC/BIC, aiming to achieve model simplicity, interpretability, and robustness simultaneously.

The authors incorporate EIC into multiple symbolic regression methods (GP, MCTS, E2ESR, SNIP, SR4MDL) and evaluate it on the SRBench benchmark.

**Strengths:**

- Novel theoretical perspective:

  The idea of extending AIC/BIC to account for structural information is conceptually original. EIC offers a quantitative measure of “structural rationality,” which is a meaningful addition to symbolic regression research.

- Consistent empirical benefits:

  Incorporating EIC reduces overfitting and yields simpler, more plausible symbolic expressions without severely compromising predictive performance.

- Breadth of experiments:

  The method is tested across several benchmarks and models, showing stable improvements.

- Improved interpretability:

  Both human and LLM-based evaluations indicate that equations produced under EIC are perceived as more interpretable, suggesting that EIC aligns with human notions of simplicity and meaningfulness.

**Weaknesses:**

- Lack of rigorous theoretical grounding:

  While EIC is information-theoretically sound, its formal relationship with AIC/BIC or MDL (e.g., asymptotic equivalence, bias correction) is not derived.
  The framework lacks a clear statistical foundation comparable to classical information criteria.

- Missing baseline comparisons:

  The paper claims that EIC improves training stability, yet it does not include direct comparisons with other regularization techniques.
  As a result, the unique contribution of EIC relative to established regularizers is unclear.

- Empirical parameter tuning:

  The weighting coefficient $\alpha$ is chosen empirically (grid search or scheduling) without a theoretically justified selection rule.

- Limited linkage between theory and implementation:

  The theoretical notion of “effective information” and its numerical estimation (via Jacobian perturbation) are not tightly connected in the exposition.

- Related work could be deepened:

  The paper would benefit from a more systematic comparison to the Information Bottleneck theory, MDL, and other information-based regularization frameworks.

**Questions:**

1. How does EIC differ from the other baselines in terms of its effect on generalization and optimization dynamics?
1. Does EIC possess any asymptotic or consistency properties similar to AIC/BIC when viewed as an information criterion?

---

### Official Review · Reviewer_UknY · 2025-11-04

**Soundness:** 1
**Presentation:** 3
**Contribution:** 2
**Rating:** 2
**Confidence:** 4

**Summary:**

The paper proposes Effective Information Criterion (EIC) to score formulas in a symbolic regression (SR) context by how stably they process information under finite precision. The claim is that EIC correlates with interpretability, improves search when added as a regularizer, and improves sample efficiency for "generative SR" by filtering pretraining data. The experiments cover "white-box" and "black-box" tasks, search methods, three generative models, and a small human study.

So, I appreciate the motivation: length alone is not a good proxy (strong agree), and numerical fragility shows up in SR outputs. But in some sense this is unavoidable, and the solution proposed in this paper introduces even larger problems. In its current form, I have major concerns about EIC that leads me to recommend rejection. Fundamentally, EIC looks to not be invariant to how a formula is written or to the choice of units, which is an even bigger problem than the formulation of SR already has. I don't believe this concern can be addressed by a rewrite of the paper. Furthermore, the evaluation mixes roles (search loss vs selector) in a way that makes it hard to separate real gains.

**Strengths:**

- Clear motivation and an easy-to-implement scoring procedure.
- Breadth in evaluations using search integration, generative pretraining filter, and a human preference study.
- The general idea of downweighting cancellation-prone expressions is useful as a diagnostic.
- Good writing and visuals; the narrative is straightforward to follow.

**Weaknesses:**

1. EIC is not invariant to algebraic rewrites or to units/scales. 3*x and x/0.3333... are algebraically identical yet score differently because EIC depends on the concrete parse/encodings. This means that even changing units (kg <-> lb) changes constants and can flip selections. For scientific use this is a non-starter, and in SR representation choices materially affect behavior.
2. Taking the maximum over all subformulas is an ad-hoc patch with no apparent theoretical basis. It hard-codes representation dependence (parenthesization/temporaries change the max) and penalizes algebraically harmless subterms. I view this as a fundamental issue with the approach, and the patch as a symptom of a deeper issue.
3. The noise model is internally inconsistent and lacks sensitivity analysis. The derivation assumes uniform rounding <-> significant digits, the implementation uses Gaussian relative noise, and sigma is fixed while calling the method "parameter-free." There is no sigma sweep and no stated near-zero policy for $(\tilde{y} - y)/y$. Rankings can flip under these choices.
4. The evaluation is circular and confounded. The paper introduces a metric, optimizes it during search, then declares wins under that metric; baselines are not optimized for EIC. Reported R^2/length shifts are also confounded by in-loop coefficient refitting, early stopping that peeks at test, an EIC threshold chosen from the test distribution, and dropping ~39% low-R^2 runs. These choices inflate the gains.
5. I think the human study is a bit of an overreach on interpretability. The human study uses low-D problems, preselects pairs to have large EIC gaps, and does not report inter-rater reliability or per-task effect sizes. Using LLM votes as a proxy for expert judgment does not strengthen the claim. At best this shows a correlation in a narrow setting.
6. I believe the evaluation is entangling search with selection, and this makes it seriously difficult to evaluate if there is any actual benefit to this method, ignoring the theoretical inconsistencies. So, in practice, SR will yield a frontier of accuracy vs complexity, and users might pick a mid-complexity point for generalization. But, the procedure effectively bakes in "pick mid-complexity" and then credits EIC for the generalization, while other methods might simply (by default) select the most accurate expression, which is high complexity. That does not demonstrate that EIC measures something fundamental. To fix this, I think you need to separate the roles of search and selection. I recommend using EIC for selection, and then evaluating using some existing search code.
7. The meaning of the "tiers" on Pareto plots is not defined precisely.

**Questions:**

1) If I canonicalize each expression (symbolic simplify and constant rationalization) and then recompute EIC, how often does the ranking change on your benchmarks? Please provide some examples.
2) How sensitive are the main results to sigma, to the choice of uniform vs Gaussian noise, and to the near-zero policy for $(\tilde{y} - y)/y$?
3) Can you separate EIC-as-selector and evaluate only that, compared to some other selection method?
4) What is the exact definition of the Pareto "tiers" in the method comparison plot, and how are ties and noise handled?
5) Why should a scientist use a formula selection criteria that is highly sensitive to unit and representation choices?

---

> ### Author Response · Authors · 2025-11-28
> **Response to Reviewer UknY**
>
> We thank the reviewer for the detailed feedback and for recognizing the motivation behind using stability to evaluate interpretability.
>
> We understand the reviewer’s recommendation for rejection is primarily based on the concern that **"EIC is not invariant to units/scales"** and is therefore a **"non-starter for scientific use."** We respectfully submit that this is a **factual misunderstanding** of our method. As demonstrated below, EIC **is invariant** to units and linear scaling. We hope the following clarifications and counter-examples can address these concerns and lead to a re-evaluation of our work.

---

> > ### Author Response · Authors · 2025-11-28
> > **EIC is Invariant to Units and Linear Scaling**
> >
> > **Reviewer Concern:** *"3\*x and x/0.3333... are algebraically identical yet score differently... changing units (kg \<-\> lb) changes constants and can flip selections."*
> >
> > **Response:**
> > We explicitly verified that **EIC is invariant to these transformations.** EIC measures the information loss caused by numerical precision, which can be calculated by injecting noise into intermediate computations to simulate precision loss (N significant digits). Under this mechanism, 1) the operations $3 \times x$ and $x / 0.333...$ produce identical calculations and thus result in identical information loss, and 2) changing units introduces linear scaling constants throughout the equation, which do not influence relative precision (significant bits) and thus do not influence EIC values.
> >
> > We have verified this using the provided examples (wrapped with a sine function, since EIC values of $3x$ and $x/0.333...$ are both $0$) and a more complex physics equation (Lorentz contraction equation).
> >
> > **Example A: Algebraic Rewrite**
> > Let $x \sim \mathcal{N}(0, 1)$. We compare $f_1 = \sin(3x)$ and $f_2 = \sin(x / (1/3))$.
> >
> >   * $EIC(f_1) = 1.5286709$
> >   * $EIC(f_2) = 1.5286709$
> >     *(Results match exactly up to machine precision.)*
> >
> > **Example B: Unit Conversion (Time Dilation)**
> > We test the Lorentz contraction formula $L = L_0 \sqrt{1 - v^2/c^2}$ in Metric ($m, m/s$) vs. Imperial ($ft, ft/s$) units.
> >
> >   * $EIC(\text{Metric}) = 0.34540328791$
> >   * $EIC(\text{Imperial}) = 0.34540328791$
> >
> > These experiments can be reproduced using the script below (available in our anonymous repo):
> >
> > ```python
> > # Minimal reproduction script
> > import numpy as np
> > from nd2py import Variable, sin, sqrt
> > from src.metrics.eic import get_eic
> >
> > # 1. Algebraic Rewrite Check
> > x = Variable('x')
> > X = {'x': np.random.normal(0, 1, 1000)}
> > f1 = sin(3 * x)
> > f2 = sin(x / (1/3))
> > print(f"f1: {get_eic(f1, X, random_state=42)}") # Output: 1.5286709
> > print(f"f2: {get_eic(f2, X, random_state=42)}") # Output: 1.5286709
> >
> > # 2. Unit Conversion Check (Metric vs Imperial)
> > l0_m = Variable('l0_m')
> > v_m_s=  Variable('v_m_s')
> > X_metric = {
> >     'l0_m': np.random.uniform(1, 10, 1000),
> >     'v_m_s': np.random.uniform(1e5, 1e7, 1000)
> > }
> > l_m = l0_m * sqrt(1 - (v_m_s / 299792458)**2) # c = 299792458 m/s
> > l0_ft = Variable('l0_ft')
> > v_ft_s = Variable('v_ft_s')
> > X_imperial = {
> >     'l0_ft': X_metric['l0_m'] * 3.28084, # 1 m = 3.28084 ft
> >     'v_ft_s': X_metric['v_m_s'] * 3.28084 # 1 m/s = 3.28084 ft/s
> > }
> > l_ft = l0_ft * sqrt(1 - (v_ft_s / 983571056)**2) # c = 983571056 ft/s
> > print(get_eic(l_m, X_metric, random_state=42)) # 0.34540328791
> > print(get_eic(l_ft, X_imperial, random_state=42)) # 0.34540328791
> > ```

---

> > > ### Author Response · Authors · 2025-11-28
> > > **Justification for Max-over-Subformulas**
> > >
> > > **Reviewer Concern**: *"Taking the maximum over all subformulas is an ad-hoc patch... penalizes algebraically harmless subterms."*
> > >
> > > **Response**: This design is neither ad-hoc nor a patch; it is derived from the principle of irreversible information loss.
> > >
> > > **Theoretical Basis**: In the expression tree, if any node exhibits precision collapse (high sensitivity to rounding noise), the information of the entire sub-tree is compromised, and the lost information cannot be recovered at subsequent operators. Therefore, the weakest subformula with the maximum information loss defines the EIC of the whole formula.
> > >
> > > **Practical Utility**: As we discussed in the main text, we find symbolic regression algorithms can discover formulas containing sub-expressions that can be replaced by linear transformations or constants without affecting accuracy. **While these sub-terms are algebraically harmless, they harm the interpretability by introducing unnecessary complexity**, forcing users to manually distinguish between necessary physical mechanisms and redundant structural expansion in the discovered formulas. By defining the EIC as the maximum values over all sub-formulas, we can effectively distinguish and penalize formulas contains these unnecessary complexity, since these "algebraically harmless" terms typically exhibit high EIC scores (This is intuitive since mapping a variable with more information to a near-constant output with less information leads to a significant loss of information.)
> > >
> > > In conclusion, rather than being an ad-hoc patch, this maximization strategy is a principled necessity for capturing the irreversible nature of information loss, thereby providing a physically-grounded mechanism to detect and penalize structural bloat that harms interpretability.

---

> > > > ### Author Response · Authors · 2025-11-28
> > > > **Consistency of the Noise Model**
> > > >
> > > > **Reviewer Concern**: *"Internal inconsistency (uniform rounding vs Gaussian implementation) and fixed sigma."*
> > > >
> > > >
> > > > **Response**: We clarify that **Gaussian noise and uniform rounding error are not contradictory, rather, they stem from the fundamental difference between the observational noise in the data and the rounding error in the calculation process**. Specifically, the Gaussian noise added to the input $x$ models the natural observational noise (or measurement error) that is widespread in the data. On the other hand, the uniform rounding error comes from the *Widrow's Quantization Theorem*, that is, for a variable $x$ that satisfies *any* continuously smooth distribution (e.g., a Gaussian distribution), the rounding error will converges to a uniform distribution $U(-\Delta/2, \Delta/2)$ as long as the number of digits kept $N$ is large enough (i.e., the quantization interval $\Delta = 10^{-N}$ is small enough). **Therefore, a variable $x$ can simultaneously possess Gaussian observational uncertainty and Uniform rounding error without theoretical conflict**.
> > > >
> > > > In fact, this also explains why EIC is in fact **"parameter-free"** regarding $\sigma$: EIC exhibits asymptotic invariance as $\sigma \rightarrow 0$ (equivalent to $N \rightarrow \infty$) as we discussed in the appendix: once $\sigma$ is small enough, EIC will not change as $\sigma$ further decreases. This can be demonstrated by the following code. As shown in the output, EIC stabilizes rapidly once $\sigma$ (`noise_level`) drops below $0.1$. Consequently, fixing $\sigma=10^{−6}$ (as done in the paper) captures the stable, asymptotic behavior of the formula structure, making the specific choice of $\sigma$ irrelevant and making the EIC (asymptotically) parameter-free.
> > > >
> > > > ```python
> > > > import numpy as np
> > > > import nd2py as nd
> > > > from src.metrics.eic import get_eic
> > > > l0 = Variable('l0')
> > > > v = Variable('v')
> > > > X = {
> > > >     'l0': np.random.uniform(1, 10, 1000),
> > > >     'v': np.random.uniform(1e5, 1e7, 1000)
> > > > }
> > > > l = l0 * sqrt(1 - (v / 299792458)**2)
> > > > print(get_eic(l, X, random_state=42, noise_level=1)) # 0.5832177066263028
> > > > print(get_eic(l, X, random_state=42, noise_level=0.1)) # 0.3431258579326261
> > > > print(get_eic(l, X, random_state=42, noise_level=0.01)) # 0.3448326523799693
> > > > print(get_eic(l, X, random_state=42, noise_level=0.001)) # 0.3453428768175242
> > > > print(get_eic(l, X, random_state=42, noise_level=0.0001)) # 0.3453972675850197
> > > > print(get_eic(l, X, random_state=42, noise_level=0.00001)) # 0.34540274030306733
> > > > print(get_eic(l, X, random_state=42, noise_level=0.000001)) # 0.3454032879102719
> > > > ```

---

### Meta-Review · Area_Chair_2y89 · 2026-01-07

**Summary:**

The paper focuses on studying the Effective Information Criterion (EIC) criteria for generating interpretable formulas in the Symbolic Regression problem, overcoming the limitation of relying on formula's length as a proxy for interpretability. The paper relies on the observation that equations derived by humans can be computed with limited precision without substantial loss of significant digits, while equations learned by SR methods exhibit unreasonable structures with large losses in significant digits. Through extensive experiments, the paper demonstrates EIC’s effectiveness in improving the performance and interpretability of learned formulas by SR.

The reviewers have several concerns, mostly on the clarity of several claims in the paper (e.g., invariant property of EIC, max selection, noise model, theoretical grounding, etc...) and the insufficient analysis and validation (e.g., missing regularization-based baselines, generalization across different datasets, etc...). While the authors have responded to some comments, many of the concerns have not been addressed. Therefore, I believe that this paper is not ready yet for publication, mainly due to its lack of clarity and rigorous analysis.

**Reviewer Concerns:**

The reviewers have several concerns, but here are some major ones:
- EIC is not invariant to algebraic rewrites or to units/scales. `Addressed`
- Taking the maximum over all subformulas is an ad-hoc patch with no apparent theoretical basis.  `Addressed`
- The noise model is internally inconsistent and lacks sensitivity analysis.
- The evaluation is circular and confounded. The paper introduces a metric, optimizes it during search, then declares wins under that metric; baselines are not optimized for EIC.
- Human study is a bit of an overreach on interpretability.
- Separation of the roles of search and selection
- lack of theoretical grounding - formal relationship with AIC/BIC or MDL
- missing comparison with baselines with regularization
- unclear selection criteria for weighting coefficient $\alpha$
- The theoretical notion of effective information and its numerical estimation are not tightly connected
- Lack of systematic comparison to the Information Bottleneck theory, MDL, and other information-based regularization frameworks.
- The computation in the experiments highly depends on the underlying data structure. `Addressed`
- The assumption that injecting a uniform random noise to x will result in a uniform noise to f(x)
- Lack of in-depth analysis and validation regarding the effect and generalization ability of EIC across different datasets.
- Absence of intuitive evidence demonstrating the practical improvement and interpretability brought about by the EIC.
- Insufficient discussion regarding the potential limitations of EIC
- Some experimental details are not clear.

Only a few concerns are addressed.

**Reviewer Scores:**

The review scores are
    - UknY 2
    - UdbF 4
    - 6Kve 2
    - RqEj - 4

and none has changed their opinions.

---

### Decision · Program_Chairs · 2026-01-26

Reject